# On the Need For Topology-Aware Generative Models for Manifold-based Defenses

**Uyeong Jang**
Department of Computer Sciences
University of Wisconsin–Madison
Madison, WI, USA
wjang@cs.wisc.edu

**Susmit Jha**
Computer Science Laboratory
SRI International
Menlo Park, CA, USA
susmit.jha@sri.com

**Somesh Jha**
Department of Computer Sciences
University of Wisconsin–Madison
Madison, WI, USA
XaiPient
Princeton, NJ, USA
jha@cs.wisc.edu

## Abstract

Machine-learning (ML) algorithms or models, especially deep neural networks (DNNs), have shown significant promise in several areas. However, researchers have recently demonstrated that ML algorithms, especially DNNs, are vulnerable to adversarial examples (slightly perturbed samples that cause misclassification). The existence of adversarial examples has hindered the deployment of ML algorithms in safety-critical sectors, such as security. Several defenses for adversarial examples exist in the literature. One of the important classes of defenses are manifold-based defenses, where a sample is "pulled back" into the data manifold before classifying. These defenses rely on the assumption that data lie in a manifold of a lower dimension than the input space. These defenses use a generative model to approximate the input distribution. In this paper, we investigate the following question: do the generative models used in manifold-based defenses need to be topology-aware? We suggest the answer is yes, and we provide theoretical and empirical evidence to support our claim.

## 1 Introduction

Machine-learning (ML) algorithms, especially deep-neural networks (DNNs), have had resounding success in several domains. However, adversarial examples have hindered their deployment in safety-critical domains, such as autonomous driving and malware detection. Adversarial examples are constructed by an adversary adding a small perturbation to a data-point so that it is misclassified. Several algorithms for constructing adversarial examples exist in the literature (Biggio et al., 2013; Szegedy et al., 2013; Goodfellow et al., 2014b; Kurakin et al., 2016a; Carlini & Wagner, 2017; Madry et al., 2017; Papernot et al., 2017). Numerous defenses for adversarial examples also have been explored (Kurakin et al., 2016b; Guo et al., 2017; Sinha et al., 2017; Song et al., 2017; Tramèr et al., 2017; Xie et al., 2017; Dhillon et al., 2018; Raghunathan et al., 2018; Cohen et al., 2019; Dubey et al., 2019).

In this paper, we focus on "manifold-based" defenses (Ilyas et al., 2017; Samangouei et al., 2018). The general idea in these defenses is to "pull back" the data point into the data manifold before classification. These defenses leverage the fact that, in several domains, natural data lies in a low-dimensional manifold (henceforth referred to as the manifold assumptions) (Zhu & Goldberg, 2009). The data distribution and hence actual manifold that the natural data lies in is usually unknown, so these defenses use a generative model to "approximate" the data distribution. Generative models attempt to learn to generate data according to the underlying data distribution. (The input to a generative model is usually random noise from a known distribution, such as Gaussian or uniform.) There are

various types of generative models in the literature, such as variational autoencoder (VAE) (Kingma & Welling, 2013), generative adversarial network (GAN) (Goodfellow et al., 2014a) and reversible generative models, e.g., real-valued non-volume preserving transform (Real NVP) (Dinh et al., 2016).

This paper addresses the following question:

> Do manifold-based defenses need to be aware of the topology of the underlying data manifold?

In this paper, we suggest the answer to this question is **yes**. We demonstrate that if the generative model does not capture the topology of the underlying manifold, it can adversely affect these defenses. In these cases, the underlying generative model is being used as an approximation of the underlying manifold. We believe this opens a rich avenue for future work on using topology-aware generative models for defense to adversarial examples.

**Contributions and Roadmap.**   We begin with a brief description of related work in Section 2. Section 3 provides the requisite mathematical background. Our main theoretical results are provided in Section 4. Informally, our result says that if the generative model is not topology-aware, it can lead to a "topological mismatch" between the distribution induced by the generative model and the actual distribution. Section 5 describes our experimental verification of our theoretical results and investigates their ramifications on a manifold-based defenses called *Invert-and-Classify (INC)* (Ilyas et al., 2017; Samangouei et al., 2018).

## 2 RELATED WORK

### 2.1 GENERATIVE MODELS

As a method for sampling high-dimensional data, generative models find applications in various fields in applied math and engineering, e.g., image processing, reinforcement learning, etc. Methods for learning data-generating distribution with neural networks include well-known examples of Variational Autoencoders (VAEs) (Kingma & Welling, 2013) and variations of Generative Adversarial Networks (GANs) (Goodfellow et al., 2014a; Radford et al., 2015; Zhao et al., 2016).

These generative models learn how to map latent variables into generated samples. The VAE is a variational Bayesian approach, so it approximates a posterior distribution over latent vectors (given training samples) by a simpler variational distribution. Similar to other variational Bayesian methods, VAE tries to minimize the Kullback–Leibler divergence between the posterior distribution and the variational distribution by minimizing the reconstruction error of the autoencoder. GANs represent another approach to learning how to transform latent vectors into samples. Unlike other approaches, the GAN learns the target distribution by training two networks – generator and discriminator – simultaneously.

In addition to generating plausible samples, some generative models construct bijective relations between latent vector and generated samples, so that the probability density of the generated sample can be estimated. Due to their bijective nature, such generative models are called to be reversible. Some examples are normalizing flow (Rezende & Mohamed, 2015), Masked Autoregressive Flow (MAF) (Papamakarios et al., 2017), Real NVP (Dinh et al., 2016), and Glow (Kingma & Dhariwal, 2018).

### 2.2 APPLICATIONS OF GENERATIVE MODELS IN ADVERSARIAL MACHINE LEARNING

The DNN-based classifier has been shown to be vulnerable to adversarial attacks (Szegedy et al., 2013; Goodfellow et al., 2014b; Moosavi-Dezfooli et al., 2016; Papernot et al., 2016; Madry et al., 2017). Several hypothesis try explaining such vulnerability (Szegedy et al., 2013; Goodfellow et al., 2014b; Tanay & Griffin, 2016; Feinman et al., 2017), and one explanation is that the adversarial examples lie far away from the data manifold. This idea leads to defenses making use of the geometry learned from the dataset – by projecting the input to the nearest point in the data manifold.

To learn a manifold from a given dataset, generative models can be exploited. The main idea is to approximate the data-generating distribution with a generative model, to facilitate searching over data manifold by searching over the space of latent vectors. The term Invert-and-Classfy (INC) was coined to describe this type of defense (Ilyas et al., 2017), and different types of generative models were tried to detect adversarial examples (Ilyas et al., 2017; Song et al., 2017; Samangouei et al., 2018). Usually, the projection is done by searching the latent vector that minimizes the geometric

distance (Ilyas et al., 2017; Samangouei et al., 2018). However, despite the promising theoretical background, all of those methods are still vulnerable (Athalye et al., 2018; Ilyas et al., 2017).

## 3 BACKGROUND

We formally describe data generation, based on the well-known manifold assumption; data lies close to a manifold whose intrinsic dimension is much lower than that of the ambient space. In our model of data generation, we provide a formal definition of *data-generating manifold* $M$ on which the data-generating distribution lies such that $M$ conforms to the manifold assumption.

### 3.1 REQUIREMENTS

Real-world data tends to be noisy, so the data does not easily correspond to an underlying manifold. We first focus on an ideal case where data is generated solely from the manifold $M$ without noise.

In the setting of classification with $l$ labels, we consider manifolds $M_1, \ldots, M_l \subset \mathbb{R}^n$ that correspond to the generation of data in each class $i \in \{1, \ldots, l\}$, respectively. We assume those manifolds are pair-wise disjoint, i.e., $M_i \cap M_j = \varnothing$ for any $i \neq j$. We set the data-generating manifold $M$ as the disjoint union of those manifolds, $M = \biguplus_{i=1}^{l} M_i$. We assume $M$ to be a compact Riemannian manifold with a volume measure $dM$ induced by its Riemannian metric. When a density function $p_M$ defined on $M$ satisfies some requirements, it is possible to compute probabilities over $M$ via $\int_{\mathbf{x} \in M} p_M(\mathbf{x}) dM(\mathbf{x})$. We call such $M$ equipped with $p_M$ an $dM$ as a data-generating manifold. We refer to Appendix A and Appendix D.1 for details about definitions and requirements on $p_M$.

In practice, data generation is affected by noise, so not all data lie on the data-generating manifold. Therefore, we incorporate the noise as an artifact of data-generation and extend the density $p_M$ on $M$ to the density $p$ on the entire $\mathbb{R}^n$ by assigning local noise densities on $M$. We consider a procedure that (1) samples a point $\mathbf{x}_o$ from $M$ first, and (2) adds a noise vector $\mathbf{n}$ to get an observed point $\hat{\mathbf{x}} = \mathbf{x}_o + \mathbf{n}$. Here, the noise $\mathbf{n}$ is a random vector sampled from a probability distribution, centered at $\mathbf{x}_o$, whose *noise density function* is $\nu_{\mathbf{x}_o}$, satisfying $\nu_{\mathbf{x}}(\mathbf{n}) = \nu_{\mathbf{x}}(\hat{\mathbf{x}} - \mathbf{x}) = p_M(\hat{\mathbf{x}} | \mathbf{x}_o = \mathbf{x})$.

### 3.2 EXTENDING DENSITY

When $M$ is equipped with a density function $p_M$ and a measure $dM$ that we can integrate over $M$, we can compute the density after random noise is added as follows.

$$p(\hat{\mathbf{x}}) = \int_{\mathbf{x} \in M} \nu_{\mathbf{x}}(\hat{\mathbf{x}} - \mathbf{x}) p(\mathbf{x}) dM(\mathbf{x}) \tag{1}$$

Since $\nu_{\mathbf{x}}(\hat{\mathbf{x}} - \mathbf{x})$ is a function on $\mathbf{x}$ when $\hat{\mathbf{x}}$ is fixed, computing this integration can be viewed as the computing expectation of a real-valued function defined on $M$. Computing such expectation has been explored in Pennec (1999). A demonstrative example is provided in Appendix B, and this extension is further discussed in Appendix D.2.

### 3.3 GENERATIVE MODELS

A generative model tries to find a statistical model for joint density $p(\mathbf{x}, y)$ (Ng & Jordan, 2002). We mainly discuss a specific type that learns a transform from one distribution $\mathcal{D}_Z$ to another target distribution $\mathcal{D}_X$. Commonly, a latent vector $\mathbf{z} \sim \mathcal{D}_Z$ is sampled from a simpler distribution, e.g., Gaussian, then a pre-trained deterministic function $G$ maps to a sample $\mathbf{x} = G(\mathbf{z})$.

Specifically, we focus on *reversible generative models* to facilitate the comparison between the density of generated samples and the target density. In this approach, the dimensions of latent vectors are set to be the same as those of the samples to be generated. Also, for a given $\mathbf{x}$, the density of $\mathbf{x}$ is estimated by the *change of variable formula* (equation (2) in Section 5.1).

### 3.4 INVERT AND CLASSIFY (INC) APPROACH FOR ROBUST CLASSIFICATION

As the data-generating manifold $M$ contains class-wise disjoint manifolds, there is a classifier $f$ on $\mathbb{R}^n$ separating these manifolds. If $f$ separates the manifolds of $M$, any misclassified point should lie out of $M$. Therefore, to change a correct classification near a manifold, any adversary would pull a sample further out of the manifold. By projecting misclassified points to the nearest manifold, we may expect the classification to be corrected by the projection. The INC method (Ilyas et al., 2017; Samangouei et al., 2018) implements this using a generative model.

The main idea of INC is to invert the perturbed sample by projecting to the nearest point on the data-generating manifold. Ideally, the data-generating manifold $M$ is accessible. For any point $(\hat{\mathbf{x}}, y)$ with $f(\hat{\mathbf{x}}) \neq y$, out-of-manifold perturbation is reduced by projecting $\hat{\mathbf{x}}$ to $\mathbf{x}^*$ on $M$. The manifold $M$

is unknown in practice. However, as $M$ is the data-generating manifold of $\mathcal{D}_X$, a generative model $G$ for $\mathcal{D}_X$ is trained to approximate $M$. Then, searching over $M$ is replaced by searching over latent vectors of $G$. More details about INC implementations are described in Section 5.1.

# 4    TOPOLOGICAL PROPERTIES OF DATA FROM GENERATIVE MODELS

In this paper, we study the significance of differences in the topological properties of the latent vector distribution and the target distribution in learning generative models. Initial information about the topology of target distribution[1] is crucial to the generative model performance. Specifically, if there is a difference in the number of connected components in the superlevel set between the target distribution and the distribution of the latent vector, then any continuous generative model $G$ cannot approximate the target distribution properly (irrespective of the training method). Due to the space limit, all proofs are presented in Appendix C.

## 4.1    TOPOLOGY OF DISTRIBUTIONS BASED ON SUPERLEVEL SETS

The data-generating manifold is a geometric shape that corresponds to the distribution. However, this manifold is not accessible in most cases and we only have indirect access via the distribution extended from it. Therefore, we consider finding a shape from the extended density so that this "shape" successfully approximates the data-generating manifold.

**$\lambda$-density superlevel set.**    We use the concept of $\lambda$-density superlevel set to capture geometric features of the density function. Simply put, for a density function $p$ and a threshold $\lambda > 0$, the *$\lambda$-density superlevel set $L_{p,\lambda}$* is the inverse image $p^{-1}[\lambda, \infty]$. Our theoretical contribution is the conditional existence of a $\lambda$-density superlevel set reflecting the topology of the data-generating manifold under proper conditions on the noise density.

**Assumptions on noise density.**    For a family of densities $\{\nu_{\mathbf{x}}\}_{\mathbf{x} \in M}$, we require the noise $\nu_{\mathbf{x}}$ to satisfy a number of assumptions. These assumptions facilitate theoretical discussion about the superlevel set reflecting the data-generating manifold. In the following definition, we denote a Euclidean ball of radius $\delta$ centered at $\mathbf{x}$ by $B_\delta(\mathbf{x})$.

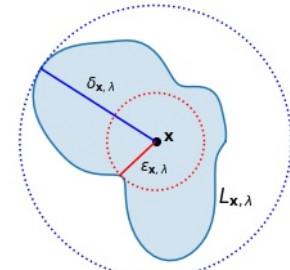

**Definition 1.** Let $\nu_{\mathbf{x}}$ be a family of noise densities.

- $\lambda$ is *small-enough* if $L_{\nu_{\mathbf{x}},\lambda}$ is nonempty for all $\mathbf{x} \in M$,

- *$\lambda$-bounding radius* $\delta_{\mathbf{x},\lambda} := \min\{\delta \mid L_{\nu_{\mathbf{x}},\lambda} \subseteq \overline{B_\delta(\mathbf{0})}\}$ is the smallest radius that $\overline{B_\delta(\mathbf{0})}$ contains $L_{\nu_{\mathbf{x}},\lambda}$. When $\max_{\mathbf{x} \in M} \delta_{\mathbf{x},\lambda}$ exists for some $\lambda$, we denote the maximum value as $\delta_\lambda$.

- *$\lambda$-guaranteeing radius* $\epsilon_{\mathbf{x},\lambda} := \max\{\epsilon \mid \overline{B_\epsilon(\mathbf{0})} \subseteq L_{\nu_{\mathbf{x}},\lambda}\}$ is the largest radius that $L_{\nu_{\mathbf{x}},\lambda}$ contains $\overline{B_\epsilon(\mathbf{0})}$. When $\min_{\mathbf{x} \in M} \epsilon_{\mathbf{x},\lambda}$ exists for some $\lambda$, we denote the minimum value as $\epsilon_\lambda$.

Figure 1: Example superlevel set $L_{\mathbf{x},\lambda}$ with $\lambda$-bounding radius $\delta_{\mathbf{x},\lambda}$ and $\lambda$-guaranteeing radius $\epsilon_{\mathbf{x},\lambda}$.

Sufficient conditions for the existence of these radii are discussed in Appendix D.3. The properties of these radii are summarized in Lemma 1. (The proof follows from Definition 1).

**Lemma 1.** Let $\nu_{\mathbf{x}}$ be a family of noise densities and let $\lambda$ be small-enough. Then,

$$\|\hat{\mathbf{x}} - \mathbf{x}\| > \delta_\lambda \implies \nu_{\mathbf{x}}(\hat{\mathbf{x}} - \mathbf{x}) < \lambda$$
$$\|\hat{\mathbf{x}} - \mathbf{x}\| \leq \epsilon_\lambda \implies \nu_{\mathbf{x}}(\hat{\mathbf{x}} - \mathbf{x}) \geq \lambda$$

whenever $\delta_\lambda$ and $\epsilon_\lambda$ exist.

Figure 1 shows an example of superlevel set $L_{\mathbf{x},\lambda}$ of noise $\nu_{\mathbf{x}}$ at a point $\mathbf{x}$ and its $\lambda$-bounding radius $\delta_{\mathbf{x},\lambda}$ and $\lambda$-guaranteeing radius $\epsilon_{\mathbf{x},\lambda}$.

---

[1]The term *topology of distributions*, refers to the topology of shapes that correspond to the distributions.

Finally, we define the continuous variation of noise densities $\nu_{\mathbf{x}}$ over changes of $\mathbf{x} \in M$. For the continuous variation, we require the continuity of both radii $\delta_{\mathbf{x},\lambda}$ and $\epsilon_{\mathbf{x},\lambda}$ as real-valued functions of $\mathbf{x} \in M$ for any fixed value of $\lambda$.

**Definition 2** (Continuously varying radii). Noise densities $\nu_{\mathbf{x}}$ have *continuously varying radii* if, for a fixed small-enough $\lambda$, both $\lambda$-bounding radius $\delta_{\mathbf{x},\lambda}$ and $\lambda$-guaranteeing radius $\epsilon_{\mathbf{x},\lambda}$ are continuous functions of $\mathbf{x} \in M$.

When noise densities have continuously varying radii, with the compactness of $M$, we can apply the extreme value theorem to guarantee the existence of both $\delta_{\lambda} = \max_{\mathbf{x} \in M} \delta_{\mathbf{x},\lambda}$ and $\epsilon_{\lambda} = \min_{\mathbf{x} \in M} \epsilon_{\mathbf{x},\lambda}$.

## 4.2 Main theorem

Our main theorem establishes, under the assumptions on noise densities from Section 4.1, the existence of a $\lambda$ such that,

- **(Inclusion)** The $\lambda$-density superlevel set $L_{p,\lambda}$ includes the data-generating manifold $M$.
- **(Separation)** The $\lambda$-density superlevel set $L_{p,\lambda}$ consists of connected components such that each component contains at most one manifold $M_i$.

**Definition 3.** Consider a data-generating manifold $M$ with density function $p_M$. For a radius $\epsilon > 0$, we define $\omega_{\epsilon}$ to be the minimum (over $\mathbf{x} \in M$) probability of sampling $\mathbf{x}' \in M$ in an $\epsilon$-ball $B_{\epsilon}(\mathbf{x})$.

$$\omega_{\epsilon} := \min_{\mathbf{x} \in M} \Pr_{\mathbf{x}' \sim p_M} \left[ \mathbf{x}' \in B_{\epsilon}(\mathbf{x}) \right]$$

**Definition 4** (Class-wise distance). Let $(X, d)$ be a metric space and let $M = \biguplus_{i=1}^{l} M_i$ be a data-generating manifold in $X$. The class-wise distance $d_{\text{cw}}$ of $M$ is defined as,

$$d_{\text{cw}} = \min_{\substack{i,j \in [l] \\ i \neq j}} \min_{\substack{\mathbf{x} \in M_i \\ \mathbf{x}' \in M_j}} d(\mathbf{x}, \mathbf{x}')$$

With the definitions above, we proved the following main theorem.

**Theorem 1.** Pick any small-enough threshold $\lambda$. Fix a value $\lambda^* \leq \omega_{\epsilon}\lambda$ and let $\delta^* = \delta_{\lambda^*}$ be the $\lambda^*$-bounding radius. If $d_{\text{cw}}$ of $M$ is larger than $2\delta^*$, then the superlevel set $L_{p,\lambda^*}$ satisfies the following properties.

- $L_{p,\lambda^*}$ contains the data-generating manifold $M$.
- Each connected component of $L_{p,\lambda^*}$ contains at most one manifold $M_i$ of class $i$.

## 4.3 Application to the generative model

We show an application of Theorem 1. We denote the target distribution by $\mathcal{D}_X$, the latent distribution by $\mathcal{D}_Z$, and the distribution of $G(\mathbf{z})$ where $\mathbf{z} \sim \mathcal{D}_Z$ by $\mathcal{D}_{G(\mathbf{z})}$. Similarly, we denote the corresponding $\lambda$-density superlevel sets of densities by $L_{\lambda}^X$, $L_{\lambda}^Z$, and $L_{\lambda}^{G(Z)}$. We assume the generative model $G$ to be continuous. Then, we get the following theorem regarding the difference between $L_{\lambda}^X$ and $L_{\lambda}^{G(Z)}$, in the number of connected components. [2]

**Theorem 2.** Let $\mathcal{D}_Z$ be a mixture of $n_Z$ multivariate Gaussian distributions, and let the data-generating manifold of $\mathcal{D}_X$ contain $n_X$ components. Let $G$ be a continuous generative model for $\mathcal{D}_X$ using latent vectors from $\mathcal{D}_Z$. Let $\lambda^*$ be the threshold value from the Theorem 1. If $n_Z < n_X$, $L_{\lambda^*}^X$ and $L_{\lambda^*}^{G(Z)}$ do not agree on the number of connected components.

We can use this theorem to deduce the need for adequate information about the target distribution when training a generative model, especially if it is used for a security-critical application, e.g., INC.

**Corollary 1.** If Theorem 2 is satisfied, there is a point $\hat{\mathbf{x}} \in \mathbb{R}^n$ such that $\hat{\mathbf{x}} \notin L_{\lambda^*}^X$ but $\hat{\mathbf{x}} \in L_{\lambda^*}^{G(Z)}$.

As a result, with density at least $\lambda^*$, $G$ generates a point $\hat{\mathbf{x}}$ that is unlikely to be generated by the target distribution. Since INC is based on generations of $G$, the INC method can output an out-of-manifold point as a solution of optimization (12).

[2]In Appendix D.4, Theorem 2 is generalized for more topological properties.

| two-moons | spirals | circles |
|---|---|---|
| $M_0 : \left\{ (x_1, x_2) \middle\| \begin{array}{l} x_1 = \cos\theta \\ x_2 = \sin\theta \end{array} \right\}$ $M_1 : \left\{ (x_1, x_2) \middle\| \begin{array}{l} x_1 = 1 - \cos\theta \\ x_2 = 1 - \sin\theta + \frac{1}{2} \end{array} \right\}$ for $\theta \in [0, \pi]$ | $M_0 : \left\{ (x_1, x_2) \middle\| \begin{array}{l} x_1 = \frac{1}{3} e^t \cos(t) \\ x_2 = \frac{1}{3} e^t \cos(t) \end{array} \right\}$ $M_1 : \left\{ (x_1, x_2) \middle\| \begin{array}{l} x_1 = \frac{1}{3} e^t \cos(t + \frac{2}{3}\pi) \\ x_2 = \frac{1}{3} e^t \sin(t + \frac{2}{3}\pi) \end{array} \right\}$ $M_2 : \left\{ (x_1, x_2) \middle\| \begin{array}{l} x_1 = \frac{1}{3} e^t \cos(t + \frac{4}{3}\pi) \\ x_2 = \frac{1}{3} e^t \sin(t + \frac{4}{3}\pi) \end{array} \right\}$ for $t \in [0, T]$ where $T = \ln\left(\frac{15}{\sqrt{2}} + 1\right)$ | $M_0 : \left\{ (x_1, x_2) \middle\| \begin{array}{l} x_1 = \cos\theta \\ x_2 = \sin\theta \end{array} \right\}$ $M_1 : \left\{ (x_1, x_2) \middle\| \begin{array}{l} x_1 = \frac{1}{2} \cos\theta \\ x_2 = \frac{1}{2} \sin\theta \end{array} \right\}$ for $\theta \in [0, 2\pi]$ |

Table 1: Parameterizations of dataset used in the experiments.

## 5 EXPERIMENTAL RESULTS

In this section, we empirically demonstrate the consequence of the two theorems and explore their implication for the INC defense. Our main goals are to provide (1) empirical support for the applicability of Theorem 2 and Corollary 1 via toy datasets, and (2) the improvement in INC performance using a class-aware generative model. The main questions and the corresponding answers are shown below.

(**Q1**) Can we experimentally verify the results of section 4.3? Specifically, can we find cases that the superlevel sets of $\mathcal{D}_X$ and $\mathcal{D}_{G(Z)}$ have different numbers of connected components?

(**Q2**) How does INC fail when the generative model is ignorant of topology information?

(**Q3**) Does the class-aware generative model improve the INC performance?

(**A1**) Theorem 2 and Corollary 1 can be verified by plotting the $\lambda$-density superlevel set. Especially, we visualize the $\lambda$-density superlevel set of $\mathcal{D}_{G(Z)}$ reflecting Theorem 2 and Corollary 1.

(**A2**) When generative model is not trained with topology information, naive INC may fail. We found out two possible reasons regarding INC failure: (1) choice of a bad initial point and (2) out-of-manifold search due to non-separation of density superlevel set.

(**A3**) The performance of INC is improved by training generative models with topology information on the target distribution. We improved the average INC performance by decreasing the error induced by projection to 30% compared to the class-ignorant counterpart.

In the rest of this section, we provide a more detailed description of our experiments. First, we briefly describe the experimental setup in Section 5.1: datasets, latent vector distributions, training method, and INC implementation. Then, Sections 5.2-5.4 describe the experimental results regarding the findings summarized above. Section 5.5 contains an additional experiment illustrating the changes of decision boundaries by INC application.

### 5.1 EXPERIMENTAL SETUP

**Datasets.** For all experiments, we use three toy datasets in $\mathbb{R}^2$: two-moons, spirals, and circles. Table 1 summarizes the parameterizations[3] of each data-generating manifold and Figure 2 shows the plots of the corresponding data-generating manifolds. To construct the training set, we first sample 1000 points uniformly from each manifold $M_i$, then each point is perturbed by isotropic Gaussian noise $\mathcal{N}(\mathbf{0}, \sigma^2 I_2)$ with $\sigma = 0.05$. Before the training, each training set is standardized by a preprocessing of Scikit-learn package.

**Latent vector distributions.** For latent vector distributions $\mathcal{D}_Z$, we prepared three different mixtures of $n_Z$ Gaussian distributions with $n_Z \in \{1, 2, 3\}$. When $n_Z = 1$, we simply use $\mathcal{N}(\mathbf{0}, I_2)$. When $n_Z = 2, 3$, we arranged $n_Z$ Gaussian distributions along a circle of radius $R = 2.5$, so that $i$-th Gaussian has mean at $\mu_i = \left( -R \sin\left(\frac{2\pi i}{n}\right), R \cos\left(\frac{2\pi i}{n}\right) \right)$ with $\sigma = 0.5$ for $n = 2$ and $\sigma = 0.3$ for $n = 3$. Then, the uniform mixtures of the arranged Gaussian are used as $\mathcal{D}_Z$. In Figure 3 (top row), we visualize the connected components corresponding to the latent vector distributions.

**Training generative models.** Our experiments mostly use the Tensorflow Probability (Dillon et al., 2017) library that contains the implementation of reversible generative models. Specifically, the Tensorflow Probability library contains an implementation of the Real NVP coupling layer that we used as a building block of our models. The default template provided by Tensorflow Probability

---

[3]The value $T$ is from the reparameterization $t = \ln\left(s/\sqrt{2} + 1\right)$ for $s \in [0, 15]$ for uniform sampling.

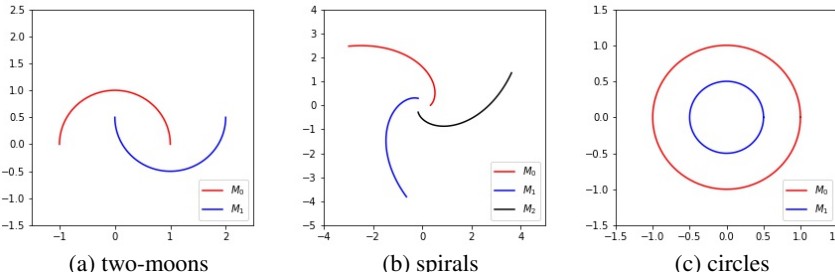

Figure 2: Data-generating manifolds used in the experiments

library was used to construct each Real NVP coupling layer with two hidden layers of 128 units. Each model uses eight coupling layers that are followed by permutations exchanging two dimensions of $\mathbb{R}^2$ except for the last coupling layer.

We describe the details of the training procedure of the generative models used in the experiments. We prepared two different types of generative models: class-ignorant and class-aware.

The *class-ignorant* type is the usual Real NVP model. This model uses the empirical estimation of negative log-likelihood over a training batch $\{\mathbf{x}_1, \ldots, \mathbf{x}_m\}$ as its training loss.

$$\ell_{\mathsf{ci}} = -\frac{1}{m} \sum_{t=1}^{m} \log(p_X(\mathbf{x}_t))$$

The density $p_X$ of $\mathcal{D}_X$ is estimated by applying *the change of variables formula*,

$$p_X(\mathbf{x}) = p_Z(\mathbf{z}) \left| \det\left( \frac{\partial G(\mathbf{z})}{\partial \mathbf{z}^T} \right) \right|^{-1} \tag{2}$$

where $p_Z$ is the density of $\mathcal{D}_Z$ and $\frac{\partial G(\mathbf{z})}{\partial \mathbf{z}^T}$ is the Jacobian of $G$ as a function from $\mathbb{R}^n$ to itself.

The *class-aware* type is the Real NVP model trained with information about the number of connected components, i.e. the number of class labels $l$. Using the number of labels, the densities $p_X$ and $p_Z$ can be decomposed as follows.

$$p_X(\mathbf{x}) = \sum_{i \in \{1, \ldots, l\}} \Pr[y = i]\, p_{X,i}(\mathbf{x})$$
$$p_Z(\mathbf{z}) = \sum_{i \in \{1, \ldots, l\}} \Pr[y = i]\, p_{Z,i}(\mathbf{z}) \tag{3}$$

where $p_{X,i}(\mathbf{x}) = p_X(\mathbf{x}|y = i)$ and each $p_{Z,i}$ is the $i$-th Gaussian component described above. Since $\Pr[y = i]$ is not generally known, the uniform distribution $\Pr[y = i] = \frac{1}{l}$ is used, where $l$ is the number of classification labels.

The main idea is class-wise training, i.e., training each $p_{X,i}$ from each $p_{Z,i}$. Applying the change of variables formula for each class $i$,

$$p_{X,i}(\mathbf{x}) = p_{Z,i}(\mathbf{z}) \left| \det\left( \frac{\partial G(\mathbf{z})}{\partial \mathbf{z}^T} \right) \right|^{-1} \tag{4}$$

Combining equations (3) and (4), we get the change of variables formula (2). We define the class-wise loss function $\ell_i$ for class-wise training as follows.

$$\ell_i = -\frac{1}{m_i} \sum_{t=1}^{m} \mathbb{1}[y_t = i]\, \log(p_{X,i}(\mathbf{x}_t))$$

where $m_i$ is the number of training samples in class $i$. Then, we train a generative model using the weighted sum of $\ell_i$ as the training loss function.

$$\ell_{\mathsf{ca}} = \sum_{i \in \{1, \ldots, l\}} \Pr[y = i]\, \ell_i$$

Each model was trained for 30,000 iterations. For each iteration, a batch of 200 random samples was chosen from two-moons and circles dataset, and a batch of 300 random samples was chosen from the spirals dataset. For the choices of latent vector distribution, we chose the mixture of $l - 1$ Gaussians for the class-ignorant type, whereas we chose the mixture of $l$ Gaussians for the class-aware type.

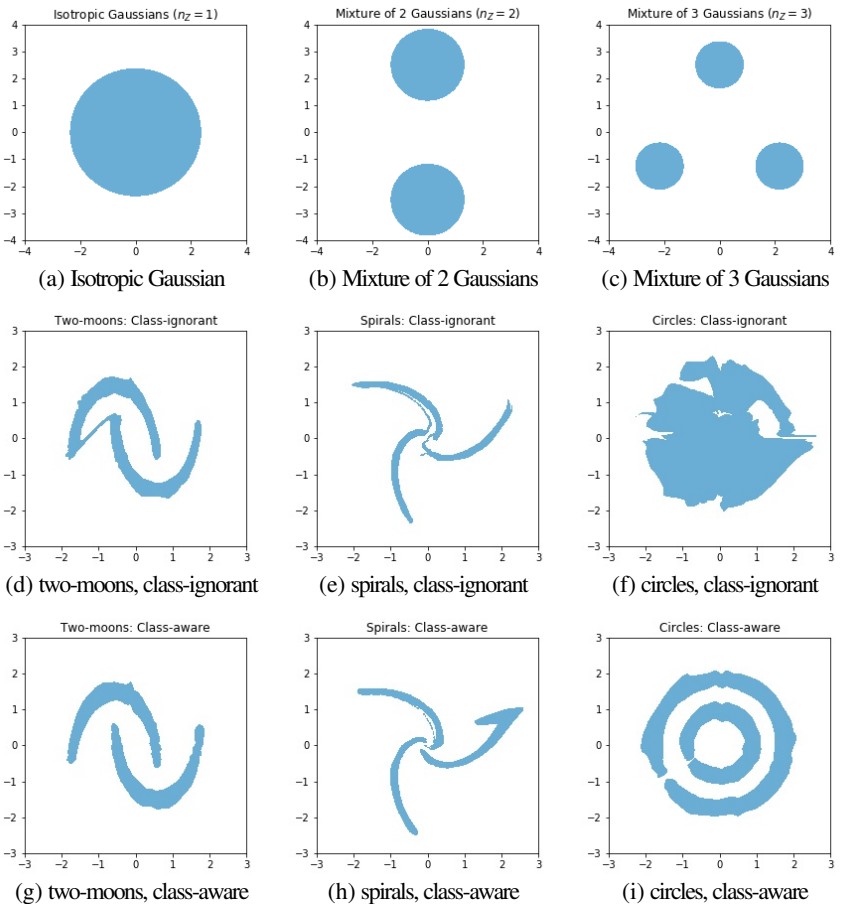

Figure 3: $\lambda$-density superlevel sets of $\mathcal{D}_Z$ and $\mathcal{D}_{G(Z)}$ with $\lambda = 0.01$. Top row: $\mathcal{D}_Z$ for $n_Z = 1, 2, 3$. Middle row: $\mathcal{D}_{G(Z)}$, class-ignorant model. Bottom row: $\mathcal{D}_{G(Z)}$, class-aware model.

## 5.2 VISUAL VERIFICATION OF THEOREMS

The goal of this section is to verify Theorem 2 and the Corollary 1 by visualizing the superlevel set reflecting the statements. Figure 3 shows the $\lambda$-density superlevel sets of densities of $\mathcal{D}_{G(Z)}$ using the same threshold $\lambda = 0.01$. The first row and the second row show the results from the class-ignorant version and those from the class-aware version, respectively. Each column corresponds to each dataset. All distributions are scaled for the standardization preprocessing before the training.

In general, superlevel set components are separated when the generative model is class-aware. On the contrary, the class-ignorant generative models introduce connections between the components, as anticipated by Corollary 1. Due to this connection, the class-ignorant generative models contain fewer connected components in their superlevel sets; this verifies Theorem 2 for our choice of $\lambda^* = 0.01$.

## 5.3 INC FAILURE DUE TO THE LACK OF INFORMATION ON THE DISTRIBUTION TOPOLOGY

We present how the non-separation of superlevel set components influences the performance of the INC. We provide two possible explanations of why the INC fails. First, the bad initialization causes a suboptimal solution on a manifold not-the-nearest to the input. Second, an artifact induced by the topological difference produces an out-of-manifold solution.

Figure 4 presents three visualized examples of INC with a class-ignorant generative model for two-moons. In each plot, the black dot is the given point $\hat{x}$, and cyan dot is the initial point from choosing $\mathbf{z}$ randomly from the latent vector distribution $-\mathcal{N}(\mathbf{0}, I_2)$, and magenta dot is the final point output by INC. All intermediate points of the optimization are plotted with dots, changing colors gradually from cyan to magenta. The training set for two-moon used in the training procedure is plotted in gray.

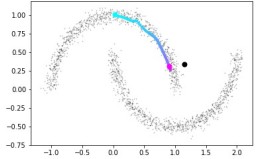 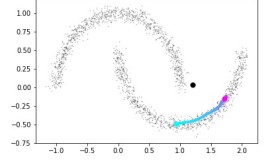 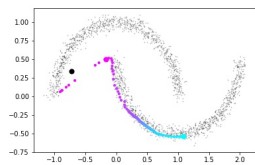

(a) INC with an ideal initialization     (b) INC with a bad initialization     (c) INC searching out of manifold

Figure 4: Successful and failed cases of INC using class-ignorant generative model of two-moon.

| Two-moons | | Spirals | | Circles | |
|---|---|---|---|---|---|
| class-ignorant | class-aware | class-ignorant | class-aware | class-ignorant | class-aware |
| 0.647 (0.666) | 0.148 (0.208) | 1.523 (1.338) | 0.443 (0.440) | 0.699 (0.491) | 0.180 (0.259) |

Table 2: Comparison of the projection errors of INC based on the class-awareness of the model.

Figure 4a is the INC optimization with an ideal start. The initial point lies in the same manifold as the manifold closest to $\hat{x}$. Then, the INC optimization searches along the manifold, converging to a point close to $\hat{x}$. Figure 4b shows a case in which INC fails because of a bad initialization. The initial point was chosen on a manifold not containing the desired solution, so the INC converged to a local optimum on the wrong manifold. Our class-aware INC performs manifold-wise initialization to circumvent this issue. Figure 4c shows that the INC failed due to an out-of-manifold search. The INC converged in a wrong manifold, and a nontrivial amount of intermediate points were out of manifold, resulting in an out-of-manifold solution (see Figure 3d).

### 5.4 INC IMPROVEMENT VIA CLASS-AWARE GENERATIVE MODEL

We demonstrate that INC performance is improved by using class-aware generative models. To measure the performance of the INC, 100 points are chosen uniformly from each manifold $M_i$. Then, each point $x$ is perturbed by $n_x$ normal to the manifold at $x$, generating 200 adversarial points $\hat{x} = x \pm r\, n_x$. For all datasets, $r = 0.2$ is used for perturbation size. We expect two types of INC to map $\hat{x}$ back to the original point $x$, as $x$ is the optimal solution to (11). We define the *projection error* of INC as $\|\text{INC}(\hat{x}) - x\|_2$, and collect the statistics of projection errors over all $\hat{x}$.

Table 2 shows the projection error statistics for two types of INC. Each pair of columns show the results on the indicated dataset. For each pair, one column shows the error of the class-ignorant INC and the other column shows that of the class-aware counterpart. Numbers in each cell are averages and standard deviations (in parenthesis) of the projection error. For any dataset, the class-aware INC achieves lower projection errors. Histograms of the projection errors are provided in Appendix E.

### 5.5 ADDITIONAL EXPERIMENTS FOR THE INC PERFORMANCE.

Finally, we present experiments to demonstrate the effect of the superlevel set discrepancy on the INC performance. First, we begin with training support vector machines (SVMs) performing classification tasks for our target distributions. For training data, we randomly sampled 1000 training points from each data-generating manifold. The baseline SVMs were intentionally ill-trained by using the high kernel coefficient $\gamma = 100$. [4] After training SVMs, we formed other classifiers by applying INC to ill-trained SVMs To explain, for each dataset, we have four types of classifiers as follows.

**(1)** Ill-trained SVM: Baseline classifier

**(2)** Ideal INC: Classifier with INC using a direct access to the data-generating manifolds

**(3)** Class-ignorant INC: Classifier with INC using a topology-ignorant generative model

**(4)** Class-aware INC: Classifier with INC with using a topology-aware generative model

We want to emphasize that direct access to the data-generating manifold is not possible in general. However, applying INC using direct access gives us an INC purely based on the geometry, so it is an ideal form of INC that should be approximated. Also, since the class-ignorant INC is affected by a bad choice of an initial point, we reduced the effect of bad initialization by sampling more initial points and taking the best solution among the projection results. For this number of initial choices, we

---

[4] In general, choosing an unnecessarily high kernel coefficient $\gamma$ causes overfitting (Chaudhuri et al., 2017), inducing decision boundary close to the training data.

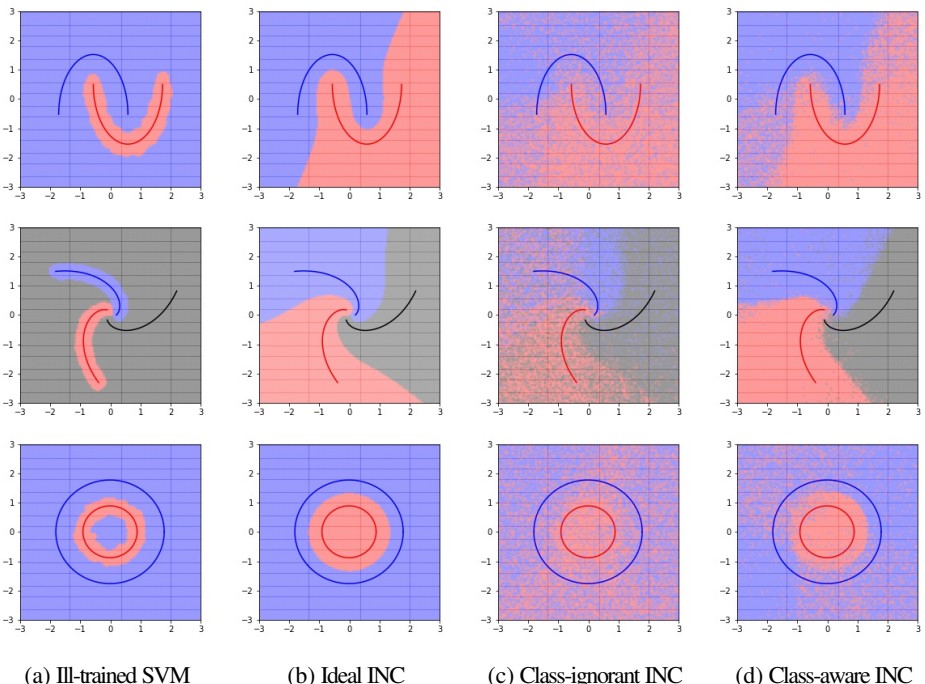

(a) Ill-trained SVM      (b) Ideal INC      (c) Class-ignorant INC      (d) Class-aware INC

Figure 5: Changes in the decision boundaries of ill-trained SVM after the INC applications.

chose as many initial points as the number of manifolds, which was exactly the same as the number of initial points for the topology-aware INC model.

To demonstrate the improvement in the robustness of the model, we visualize the effect by depicting the decision boundary of each classifier. Specifically, we form a $300 \times 300$ grid on the domain of $[-3, 3] \times [-3, 3]$ and compute the classification result. The depicted decision boundaries are presented in Figure 5. Each row corresponds to each dataset: two moons, spirals, and circles, respectively. Each column corresponds to classifiers 1-4 described above, from the first column to the fourth column, respectively. From Figure 5, it is visually evident that the class-aware INC models provide more proper approximations to the ideal INC model compared to the class-ignorant INC models.

## 6 CONCLUSION

We theoretically and experimentally discussed the necessity of topology awareness in the training of generative models, especially in security-critical applications. A continuous generative model is sensitive to the topological mismatch between the latent vector distribution and the target distribution. Such mismatch leads to potential problems with manifold-based adversarial defenses utilizing generative models such as INC. We described two cases in which the INC failed: the bad initialization and the artifacts from the topological difference. We experimentally verified that topology-aware training effectively prevented these problems, thereby improving the effectiveness of generative models in manifold-based defense. After topology-aware training of generative models, the INC projection errors represented 30% of the errors of the topology-ignorant INC.

## 7 ACKNOWLEDGEMENT

Dr. Susmit Jha and Uyeong Jang's internship at SRI International were supported in part by U.S. National Science Foundation (NSF) grants #1740079, #1750009, U.S. Army Research Laboratory Cooperative Research Agreement W911NF-17-2-0196, and DARPA Assured Autonomy under contract FA8750-19-C-0089. The views, opinions and/or findings expressed are those of the author(s) and should not be interpreted as representing the official views or policies of the Department of Defense or the U.S. Government. This work is partially supported by Air Force Grant FA9550-18-1-0166, the National Science Foundation (NSF) Grants CCF-FMitF-1836978, SaTC-Frontiers-1804648 and CCF-1652140 and ARO grant number W911NF-17-1-0405.

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

# A MATHEMATICAL BACKGROUND

## A.1 GENERAL TOPOLOGY

We introduce definitions and theorems related to general topology appeared in the paper. For more details, all the definitions and theorems can be found in Munkres (2000).

**Definitions in general topology.** We first provide the precise definitions of the terms we brought from the general topology.

**Definition 5** (Topological space). A *topology* on a set $X$ is a collection $\mathcal{T}$ of subsets of $X$ having the following properties.

1. $\varnothing$ and $X$ are in $\mathcal{T}$.

2. The union of the elements of any subcollection of $\mathcal{T}$ is in $\mathcal{T}$.

3. The intersection of the elements of any finite subcollection of $\mathcal{T}$ is in $\mathcal{T}$.

A set $X$ for which a topology $\mathcal{T}$ has been specified is called a *topological space*.

For example, a collection of all open sets in $\mathbb{R}^n$ is a topology, thus $\mathbb{R}^n$ is a topological space. If a topology can be constructed by taking arbitrary union and a finite number of intersections of a smaller collection $\mathcal{B}$ of subsets of $X$, we call $\mathcal{B}$ is a basis of the topology.

Pick a metric $d$ in $\mathbb{R}^n$ and consider $\mathcal{B}$ a set of all open balls in $\mathbb{R}^n$ using the metric $d$. The topology of $\mathbb{R}^n$ can be constructed by taking $\mathcal{B}$ as a basis. When this construction is possible, metric $d$ is said to *induce the topology*.

**Definition 6** (Metrizable space). If $X$ is a topological space, $X$ is said to be *metrizable* if there exists a metric $d$ on the set $X$ that induces the topology of $X$. A *metric space* is a metrizable space $X$ together with a specific metric $d$ that gives the topology of $X$.

Since $\mathbb{R}^n$ is equipped with Euclidean metric that induces its topology, $\mathbb{R}^n$ is metrizable.

**Continuity and the extreme value theorem.** Let $X$ and $Y$ be topological spaces. In the field of general topology, a function $f : X \to Y$ is said to be *continuous*, if for any subset $V$ open in $Y$, its inverse image $f^{-1}(V)$ is open in $X$. Moreover, if $f$ is a continuous bijection whose inverse is also continuous, $f$ is called a *homeomorphism*. The notion of homeomorphism is important as it always preserves topological property, e.g., connectedness, compactness, etc., and this will be used in the further generalization of Theorem 2.

Here, we only introduce the generalized statement of extreme value theorem.

**Theorem 3** (Extreme value theorem). Let $f : X \to Y$ be continuous, where $Y$ is an ordered set. If $X$ is compact, then there exist points $\underline{\mathbf{x}}$ and $\overline{\mathbf{x}}$ in $X$ such that $f(\underline{\mathbf{x}}) \leq f(\mathbf{x}) \leq f(\overline{\mathbf{x}})$ for every $\mathbf{x} \in X$.

Specifically, if a manifold $M$ is a compact subset in $\mathbb{R}^n$, we may use $X = M$ and $Y = \mathbb{R}$.

**Normal space and Urysohn's lemma.** The Urysohn's lemma was used to prove the Corollary 1. We first introduce the notion of normal space.

**Definition 7** (Normal space). Let $X$ be a topological space that one-point sets in $X$ are closed. Then, $X$ is *normal* if for each pair $A$, $B$ of disjoint closed sets of $X$, there exist disjoint open sets containing $A$ and $B$, respectively.

Urysohn's lemma is another equivalent condition for a space to be normal.

**Theorem 4** (Urysohn's lemma). Let $X$ be a normal topological space; let $A$ and $B$ be disjoint closed subsets in $X$. Let $[a, b]$ be a closed interval in the real line. Then there exists a continuous map

$$f : X \longrightarrow [a, b]$$

such that $f(\mathbf{x}) = a$ for every $\mathbf{x}$ in $A$, and $f(\mathbf{x}) = b$ for every $\mathbf{x}$ in $B$.

To apply this lemma to $\mathbb{R}^n$, we only need the following theorem.

**Theorem 5.** Every metrizable space is normal.

Since $\mathbb{R}^n$ is metrizable, it is a normal space by Theorem 5. Therefore, we can apply Urysohn's lemma to any pair of disjoint subsets in $\mathbb{R}^n$, to show the existence of a continuous map $f : X \to [0, 1]$.

## A.2 Differential geometry

We provide the definitions from differential geometry (Lee, 2003) used in the paper.

**Manifold and tangent space.** Formally, topological manifold is defined as follows.

**Definition 8** (Manifold). Suppose $M$ is a topological space. We say $M$ is a topological manifold of dimension $k$ if it has the following properties.

1. For any pair of distinct points $\mathbf{x}_1, \mathbf{x}_2 \in M$, there are disjoint open subsets $U_1, U_2 \subset M$ such that $\mathbf{x}_1 \in U$ and $\mathbf{x}_2 \in V$.

2. There exists a countable basis for the topology of $M$.

3. Every point has a neighborhood $U$ that is homeomorphic to an open subset $\tilde{U}$ of $\mathbb{R}^k$.

There are different ways to define tangent space of $k$-dimensional manifold $M$. Informally, it can be understood as *geometric tangent space* to $M \subset \mathbb{R}^n$ at a point $\mathbf{x} \in M$, which is a collection of pairs $(\mathbf{x}, \mathbf{v})$ where $\mathbf{v}$ is a vector tangentially passing through $\mathbf{x}$. Here we put a more formal definition of tangent space. Consider a vector space $C^\infty(M)$, a set of smooth functions on $M$.

**Definition 9** (Tangent space). Let $\mathbf{x}$ be a point of a smooth manifold $M$. A linear map $X : C^\infty(M) \to \mathbb{R}$ is called a *derivation at* $\mathbf{x}$ if it satisfies

$$X(fg) = f(\mathbf{x})Xg + g(\mathbf{x})Xf$$

for all $f, g \in C^\infty(M)$.

The set of all derivations of $C^\infty(M)$ at $\mathbf{x}$ forms a vector space called the *tangent space* to $M$ at $\mathbf{x}$, and is denoted by $T_\mathbf{x}(M)$.

**Riemannian metric.** As tangent space $T_\mathbf{x}(M)$ is a vector space for each $\mathbf{x} \in M$, we can consider a inner product $g_{bfx}$ defined on $T_\mathbf{x}(M)$.

**Definition 10** (Riemannian metric). A *Riemannian metric* $g$ on a smooth manifold $M$ is a smooth collection of inner products $g_\mathbf{x}$ defined for each $T_\mathbf{x}(M)$. The condition for smoothness of $g$ is that, for any smooth vector fields $\mathcal{X}$, $\mathcal{Y}$ on M, the mapping $\mathbf{x} \mapsto g_\mathbf{x}(\mathcal{X}|_\mathbf{x}, \mathcal{Y}|_\mathbf{x})$.

A manifold $M$ equipped with a Riemannian metric $g$ is called a Riemannian manifold.

## B Examples

**Computing density $p_M$ over a Riemannian manifold $M$.** This section presents example computations of the probability computations from Section D.1 and Section 3.2 As a concrete example of computing density over a manifold, we use the following simple manifolds, so called *two-moons* in $\mathbb{R}^2$.

$$M_0 = \left\{ (x_1, x_2) \middle| \begin{array}{l} x_1 = \cos\theta \\ x_2 = \sin\theta \end{array} \text{ for } \theta \in [0, \pi] \right\}$$

$$M_1 = \left\{ (x_1, x_2) \middle| \begin{array}{l} x_1 = 1 - \cos\theta \\ x_2 = 1 - \sin\theta + \frac{1}{2} \end{array} \text{ for } \theta \in [0, \pi] \right\}$$

We take $M = M_0 \cup M_1$ as our example manifold. Figure 6a shows the manifold of two-moons dataset plotted in different colors: $M_0$ in red and $M_1$ in blue.

First recall the following equation (equation (8) from the Section D.1).

$$\int_{\mathbf{x} \in M} p_M(\mathbf{x}) dM(\mathbf{x}) = \int_{\mathbf{u} \in D} p_M(X(\mathbf{u})) \sqrt{\left| \det[g_{X(\mathbf{u})}] \right|} d\mathbf{u}$$

where $[g_{X(\mathbf{u})}]$ is the $k \times k$ matrix representation of the inner product $g_{X(\mathbf{u})}$ at $X(\mathbf{u}) \in M$.

Especially, when a manifold in $\mathbb{R}^n$ is of dimension 1, i.e., parameterized curve $\gamma : [a, b] \to \mathbb{R}^n$, the integration (8) can be written in simpler way.

$$\int_{\mathbf{x} \in M} p_M(\mathbf{x}) dM(\mathbf{x}) = \int_{t=a}^{b} p_M(\gamma(t)) \|\gamma'(t)\| dt \tag{5}$$

where $\gamma'(t)$ is the $n$-dimensional velocity vector at $t \in [a, b]$.

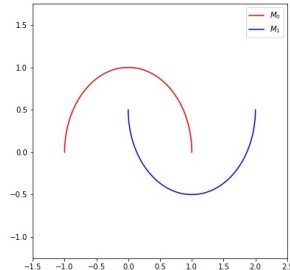
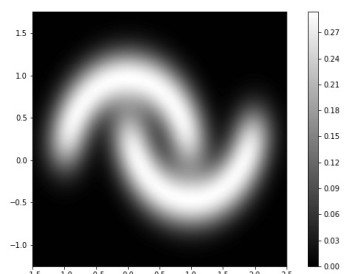

(a) Plot of the two-moons manifold in $\mathbb{R}^2$

(b) Extended density function over $\mathbb{R}^2$ from the two-moons dataset

Figure 6: Density extension example from two-moons manifold.

Let $p_M$ be a probability density function defined on $M$. As $M$ is composed of two disjoint manifolds $M_0$ and $M_1$, we consider conditional densities $p_0, p_1$ as follows.

$$p_0(\mathbf{x}) = p_M(\mathbf{x} \,|\, \mathbf{x} \in M_0) = \frac{p_M|_{M_0}(\mathbf{x})}{\Pr[\mathbf{x} \in M_0]}$$

$$p_1(\mathbf{x}) = p_M(\mathbf{x} \,|\, \mathbf{x} \in M_1) = \frac{p_M|_{M_1}(\mathbf{x})}{\Pr[\mathbf{x} \in M_1]} \tag{6}$$

Here, $p_M|_{M_0}$ and $p_M|_{M_1}$ represent the density function $p_M$ with its domain restricted to $M_0$ and $M_1$, respectively. By our definition of data-generating manifolds, $\Pr[\mathbf{x} \in M_i]$ corresponds to the probability of data generation for class $i$, i.e. $\Pr[y = i]$. For a concrete example of such density, uniform density for each manifold $M_i$ can be defined as $p_i(\mathbf{x}) = \frac{1}{\pi}$ for all $\mathbf{x} \in M_i$.

Note that each manifold has parameterized curves in $\mathbb{R}^2$,

$$\gamma_0 : \theta \mapsto (\cos\theta, \sin\theta)$$
$$\gamma_1 : \theta \mapsto (1 - \cos\theta, 1 - \sin\theta + 0.5)$$

with constant speed $\|\gamma_0'(\theta)\| = \|\gamma_1'(\theta)\| = 1$ at all $\theta \in [0, \pi]$. Therefore, from equation (5),

$$\int_{\mathbf{x} \in M_0} p_M|_{M_0}(\mathbf{x}) dM_0(\mathbf{x}) = \int_{\theta=0}^{\pi} p_M(\gamma_0(\theta)) d\theta$$

$$\int_{\mathbf{x} \in M_0} p_M|_{M_1}(\mathbf{x}) dM_1(\mathbf{x}) = \int_{\theta=0}^{\pi} p_M(\gamma_1(\theta)) d\theta \tag{7}$$

For any measurable subset $A \subseteq M$, the probability for an event that $\mathbf{x}$ is in $A$ can be computed as follows.

$$\begin{aligned}
\Pr[\mathbf{x} \in A] &= \int_{\mathbf{x} \in A \subseteq M} p_M(\mathbf{x}) dM(\mathbf{x}) \\
&= \int_{\mathbf{x} \in A \cap M_0} p_M|_{M_0}(\mathbf{x}) dM_0(\mathbf{x}) + \int_{\mathbf{x} \in A \cap M_1} p_M|_{M_1}(\mathbf{x}) dM_1(\mathbf{x}) \\
&= \int_{\substack{\theta \in [0,\pi] \\ \gamma_0(\theta) \in A}} p_M|_{M_0}(\gamma_0(\theta)) d\theta + \int_{\substack{\theta \in [0,\pi] \\ \gamma_1(\theta) \in A}} p_M|_{M_1}(\gamma_1(\theta)) d\theta \quad (\because (7)) \\
&= \Pr[\mathbf{x} \in M_0] \int_{\substack{\theta \in [0,\pi] \\ \gamma_0(\theta) \in A}} p_0(\gamma_0(\theta)) d\theta \\
&\quad + \Pr[\mathbf{x} \in M_1] \int_{\substack{\theta \in [0,\pi] \\ \gamma_1(\theta) \in A}} p_1(\gamma_1(\theta)) d\theta \quad (\because (6)) \\
&= \frac{1}{\pi} \left( \Pr[\mathbf{x} \in M_0] \int_{\substack{\theta \in [0,\pi] \\ \gamma_0(\theta) \in A}} 1 d\theta + \Pr[\mathbf{x} \in M_1] \int_{\substack{\theta \in [0,\pi] \\ \gamma_1(\theta) \in A}} 1 d\theta \right)
\end{aligned}$$

We can briefly check all the requirements (**R1**), (**R2**), and (**R3**). The computation of $\Pr[\mathbf{x} \in A]$ is based on (**R1**), so (**R1**) is satisfied trivially. Also, $p_M$ is a function defined only on $M$, thus (**R2**) is clear, i.e. $\text{supp}(p_M) = \{\mathbf{x} \in \mathbb{R}^n \,|\, p(\mathbf{x}) > 0\} \subseteq M$. To check (**R3**), when $A = M_i$, computing this integration will result in the exact probability $\Pr[\mathbf{x} \in M_i] = \Pr[y = i]$, so when $A = M$, computing the integration will result in $\Pr[y = 0] + \Pr[y = 1] = 1$, as desired in the requirements.

**Extending density to $\mathbb{R}^n$.** We extend the domain to $\mathbb{R}^n$ for the example of two-moon. In Section 3, we defined the noise density function to satisfy the following requirement.

(**R0**) The *translated noise density function*, $\nu_\mathbf{x}(\hat{\mathbf{x}} - \mathbf{x})$, is the density of noise $\mathbf{n} = \hat{\mathbf{x}} - \mathbf{x}$ being chosen for a given $\mathbf{x}$. Given $\mathbf{x}_o = \mathbf{x}$, since adding noise $\mathbf{n}$ is the only way to generate $\hat{\mathbf{x}}$ by perturbing $\mathbf{x}_0$, $p(\hat{\mathbf{x}}|\mathbf{x}_o = \mathbf{x})$ is equal to $\nu_\mathbf{x}(\mathbf{n})$.

Under a proper noise density function, We show an example construction of the density extended from $M$ satisfying the requirement (**R0**). For simplicity, we choose isotropic Gaussian distribution, $\mathcal{N}(0, \sigma^2 I)$ with the standard deviation $\sigma$ for each dimension as the noise density function $\nu_\mathbf{x}$ for all $\mathbf{x} \in M$. Such noise density $\nu_\mathbf{x}$ defined in $\mathbb{R}^n$ can be written as follows.

$$\nu_\mathbf{x}(\mathbf{n}_\mathbf{x}) = \frac{1}{\sqrt{2\pi\sigma^2}} \exp\left(-\frac{\|\mathbf{n}_\mathbf{x}\|_2^2}{2\sigma^2}\right)$$

By putting $\mathbf{n}_\mathbf{x} = \hat{\mathbf{x}} - \mathbf{x}$ to density equation above,

$$p(\hat{\mathbf{x}}) = \int_{\mathbf{x} \in M} \frac{1}{\sqrt{2\pi\sigma^2}} \exp\left(-\frac{\|\hat{\mathbf{x}} - \mathbf{x}\|_2^2}{2\sigma^2}\right) p_M(\mathbf{x}) dM(\mathbf{x})$$

Specifically, We assume an isotropic Gaussian distribution with $\sigma = 0.05$ as the noise density $\nu_\mathbf{x}$ for all $\mathbf{x} \in M$.

By the equation (1), we have the following computation of density on $\hat{\mathbf{x}}$.

$$
\begin{aligned}
p(\hat{\mathbf{x}}) &= \int_{\mathbf{x} \in M} \nu_\mathbf{x}(\hat{\mathbf{x}} - \mathbf{x}) p_M(\mathbf{x}) dM(\mathbf{x}) \\
&= \int_{\mathbf{x} \in M_0} \nu_\mathbf{x}(\hat{\mathbf{x}} - \mathbf{x}) p_M|_{M_0}(\mathbf{x}) dM_0(\mathbf{x}) + \int_{\mathbf{x} \in M_1} \nu_\mathbf{x}(\hat{\mathbf{x}} - \mathbf{x}) p_M|_{M_1}(\mathbf{x}) dM_1(\mathbf{x}) \\
&= \int_{\theta=0}^{\pi} \nu_\mathbf{x}(\hat{\mathbf{x}} - \mathbf{x}) p_M|_{M_0}(\gamma_0(\theta)) d\theta + \int_{\theta=0}^{\pi} \nu_\mathbf{x}(\hat{\mathbf{x}} - \mathbf{x}) p_M|_{M_1}(\gamma_1(\theta)) d\theta \quad (\because (5)) \\
&= \Pr[\mathbf{x} \in M_0] \int_{\theta=0}^{\pi} \nu_\mathbf{x}(\hat{\mathbf{x}} - \mathbf{x}) p_0(\gamma_0(\theta)) d\theta \\
&\quad + \Pr[\mathbf{x} \in M_1] \int_{\theta=0}^{\pi} \nu_\mathbf{x}(\hat{\mathbf{x}} - \mathbf{x}) p_1(\gamma_1(\theta)) d\theta \quad (\because (6)) \\
&= \frac{1}{\pi\sqrt{2\pi\sigma^2}} \left[ \Pr[\mathbf{x} \in M_0] \int_{\theta=0}^{\pi} \exp\left(-\frac{\|\hat{\mathbf{x}} - \mathbf{x}\|_2^2}{2\sigma^2}\right) d\theta \right. \\
&\quad \left. + \Pr[\mathbf{x} \in M_1] \int_{\theta=0}^{\pi} \exp\left(-\frac{\|\hat{\mathbf{x}} - \mathbf{x}\|_2^2}{2\sigma^2}\right) d\theta \right]
\end{aligned}
$$

We can also check that the requirement (**R0**) is satisfied by the construction; our construction (equation (1)) is based on (**R0**). The computed density is shown in Figure 6b.

## C  PROOFS

In this section, we provide the proofs for statements that appeared in Section 4.

### C.1  PROOF OF THEOREM 1

To begin with, pick a value $\lambda$ such that the $\lambda$-density superlevel set $L_{\nu_\mathbf{x}, \lambda}$ is nonempty for all $\mathbf{x} \in M$. As we use noise densities $\nu_\mathbf{x}$ described in Section 4.1, it is safe to assume that both $\lambda$-bounding radius $\delta_\lambda = \max_{\mathbf{x} \in M} \delta_{\mathbf{x}, \lambda}$ and $\lambda$-guaranteeing radius $\epsilon_\lambda = \min_{\mathbf{x} \in M} \epsilon_{\mathbf{x}, \lambda}$ exist.

Then, we can prove that, with a proper choice of threshold $\lambda$, the $\lambda$-density superlevel set includes the data-generating manifold.

**Lemma 2.** Assume that noise densities have radii in Definition 1 for all $\mathbf{x} \in M$ and a small enough $\lambda > 0$. Then, for any $\mathbf{x} \in M$, the density $p(\mathbf{x})$ is at least $\omega_\epsilon \lambda$, i.e. $p(\mathbf{x}) \geq \omega_\epsilon \lambda$, where $\epsilon = \epsilon_\lambda$.

*Proof.* By Lemma 1,

$$
\begin{aligned}
\mathbf{x}' \in B_\epsilon(\mathbf{x}) &\iff \mathbf{x} \in B_\epsilon(\mathbf{x}') = B_{\epsilon_\lambda}(\mathbf{x}') \quad (\because \epsilon = \epsilon_\lambda) \\
&\implies \nu_{\mathbf{x}'}(\mathbf{x} - \mathbf{x}') \geq \lambda
\end{aligned}
$$

Then, we can lower bound the density $p_M(\mathbf{x})$ as follows.

$$
\begin{aligned}
p(\mathbf{x}) &= \int_{\mathbf{x}' \in M} \nu_{\mathbf{x}'}(\mathbf{x} - \mathbf{x}') p_M(\mathbf{x}') dM(\mathbf{x}') \\
&\geq \int_{\mathbf{x}' \in M \cap B_\epsilon(\mathbf{x})} \nu_{\mathbf{x}'}(\mathbf{x} - \mathbf{x}') p_M(\mathbf{x}') dM(\mathbf{x}') \\
&\geq \lambda \int_{\mathbf{x}' \in M \cap B_\epsilon} p_M(\mathbf{x}') dM(\mathbf{x}') \\
&= \lambda \Pr_{\mathbf{x}' \in M} [\mathbf{x}' \in B_\epsilon(\mathbf{x})] \\
&\geq \omega_\epsilon \lambda
\end{aligned}
$$

$\square$

This lemma shows that the thresholding the extended density $p$ with threshold $\lambda^* \leq \omega_\epsilon \lambda$ guarantees the superlevel set to include the entire manifold $M$.

**Corollary 2.** For any threshold $\lambda^* \leq \omega_\epsilon \lambda$, the corresponding $\lambda^*$-density superlevel set $L_{p,\lambda^*}$ of the extended density $p$ includes the data-generating manifold $M$.

Similarly, we show that, with a proper choice of threshold $\lambda$, each connected component of $\lambda$-density superlevel set contains at most one manifold.

**Lemma 3.** Assume a family of noise densities satisfies the assumptions of Section 4.1. Let $\lambda > 0$ be a value such that the $\lambda$-density superlevel set $L_{\nu_\mathbf{x}, \lambda}$ is nonempty for any $\mathbf{x} \in M$. Also, let $\delta = \delta_\lambda$ be the maximum $\lambda$-bounding radius over $M$. Then, for any $\hat{\mathbf{x}} \notin N_\delta(M)$, the extended density value is smaller than $\lambda$, i.e. $p(\hat{\mathbf{x}}) < \lambda$.

*Proof.* By Lemma 1,

$$
\begin{aligned}
\hat{\mathbf{x}} \notin N_\delta(M) &\iff \hat{\mathbf{x}} \notin B_\delta(\mathbf{x}) = B_{\delta_\lambda}(\mathbf{x}) \text{ for any } \mathbf{x} \in M \quad (\because \delta = \delta_\lambda) \\
&\implies \nu_\mathbf{x}(\hat{\mathbf{x}} - \mathbf{x}) < \lambda \text{ for any } \mathbf{x} \in M
\end{aligned}
$$

Then, we can upper bound the density $p(\hat{\mathbf{x}})$ as follows.

$$
\begin{aligned}
p(\hat{\mathbf{x}}) &= \int_{\mathbf{x} \in M} \nu_\mathbf{x}(\hat{\mathbf{x}} - \mathbf{x}) p_M(\mathbf{x}) dM(\mathbf{x}) \\
&< \lambda \int_{\mathbf{x} \in M} p_M(\mathbf{x}) dM(\mathbf{x}) \quad (\because \hat{\mathbf{x}} \notin N_{\delta_\lambda}(M)) \\
&= \lambda
\end{aligned}
$$

$\square$

This lemma says that the $\lambda$-density superlevel set is included by the $\delta$-neighborhood $N_\delta(M)$ of the data-generating manifold $M$.

Now, we can deduce the following main result.

**Theorem 1.** Pick any $\lambda^* \leq \omega_\epsilon \lambda$ threshold value satisfying the Corollary 2. If the class-wise distance of data-generating manifold is larger than $2\delta^*$ where $\delta^* = \delta_{\lambda^*}$(the $\lambda^*$-bounding radius), then the superlevel set $L_{p,\lambda^*}$ satisfies the followings.

- $L_{p,\lambda^*}$ contains the data-generating manifold $M$.

- Each connected component of $L_{p,\lambda^*}$ contains at most one manifold $M_i$ of class $i$.

*Proof.* The first property is a direct application of Corollary 2 for $\lambda^* = \omega_\epsilon \lambda$.

For the second property, since the class-wise distance of $M$ is larger than $2\delta^*$, the $\delta^*$-neighborhood of manifolds are pairwise disjoint, i.e. $N_{\delta^*}(M_i) \cap N_{\delta^*}(M_j) = \varnothing$ for each $i \neq j$. Therefore, $N_{\delta^*}(M)$ has exactly $k$ connected components $N_i = N_{\delta^*}(M_i)$'s.

By Lemma 3, $\delta^*$-neighborhood $N_{\delta^*}(M)$ contains the superlevel set $L_{p,\lambda^*}$, thus each connected component of $L_{p,\lambda^*}$ is in exactly one of $N_i$'s. Since $M$ is contained in $L_{p,\lambda^*}$, each $M_i$ is contained in some connected component $C$ of $L_{p,\lambda^*}$ which is in $N_i$. Then, for any $j \neq i$, $M_j \notin C \subset N_i$, since $M_j$ is in $N_j$ which is disjoint to $N_i$. Therefore, if a connected component $C$ contains a manifold $M_i$, then it cannot contain any other manifold. $\square$

## C.2 Proofs for Section 4.3

**Theorem 2.** Let $\mathcal{D}_Z$ be a mixture of $n_Z$ multivariate Gaussian distributions, and let $\mathcal{D}_X$ be the target distribution from a data-generating manifold with $n_X$ manifolds. Let $G$ be a continuous generative model for $\mathcal{D}_X$ using latent vectors from $\mathcal{D}_Z$. Assume the Theorem 1 is satisfied, and let $\lambda^*$ be the threshold value from the Theorem 1. If $n_Z < n_X$, $L_{\lambda^*}^X$ and $L_{\lambda^*}^{G(Z)}$ do not agree on the number of connected components.

*Proof.* Since $L_{\lambda^*}^X$ is the results of Theorem 1, the number of connected components of $L_{\lambda^*}^X$ is at least $n_X$.

However, since $\mathcal{D}_Z$ is a mixture of Gaussians, for any value of $\lambda$ (including the special case $\lambda = \lambda^*$), $L_\lambda^Z$ can never have more than $n_Z$ connected components. Since $G$ is continuous, it preserves the number of connected components, thus $L_{\lambda^*}^{G(Z)} = G(L_{\lambda^*}^Z)$ has at most $n_Z$ connected components. As $n_Z < n_X$, $L_{\lambda^*}^X$ and $L_{\lambda^*}^{G(Z)}$ can never agree on the number of connected components. $\square$

**Corollary 1.** If Theorem 2 is satisfied, there is a point $\hat{\mathbf{x}} \in \mathbb{R}^n$ such that $\hat{\mathbf{x}} \notin L_{\lambda^*}^X$ but $\hat{\mathbf{x}} \in L_{\lambda^*}^{G(Z)}$.

*Proof.* Since $n_Z < n_X$, there exists a connected components $\hat{C}$ of $L_{\lambda^*}^{G(Z)}$ containing at least two connected components of $S_{\lambda^*}^X$. Without loss of generality, assume $\hat{C}$ contains exactly two connected components $C$ and $C'$. By definition, $\lambda$-superlevel set is a closed set, so $C$ and $C'$ are disjoint closed sets. In Euclidean space $\mathbb{R}^n$, the Urysohn's lemma tells us that for any disjoint pair of closed sets $A, A'$ in $\mathbb{R}^n$, there is a continuous function $f$ such that $f|_A(\mathbf{x}) = 0$ and $f|_{A'}(\mathbf{x}) = 1$ for any $\mathbf{x} \in \mathbb{R}^n$. Especially, when $A = C$ and $A' = C'$, there exists a continuous function $f$ such that,

- $f(\mathbf{x}) = 0$ for all $\mathbf{x}$ in $C$
- $f(\mathbf{x}) = 1$ for all $\mathbf{x}$ in $C'$

Consider $S = f^{-1}(\frac{1}{2})$ which is a separating plane separating $C$ and $C'$. If $\hat{C} \cap S = \varnothing$, then $\hat{C} \cap S = f^{-1}[0, \frac{1}{2})$ and $\hat{C} \cap S = f^{-1}(\frac{1}{2}, 1]$ will be two open set in subspace $\hat{C}$, whose union is $\hat{C}$. This implies that $\hat{C}$ is disconnected, which is a contradiction. Therefore, $\hat{C} \cap S$ should be nonempty, and any point $\mathbf{x}$ in $\hat{C} \cap S$ is not in $L_{\lambda^*}^X$. $\square$

# D Further discussions

## D.1 Computing density over a data-generating manifold

When $M$ is a Riemannian manifold equipped with a Riemannian metric $g$, we can compute probabilities over $M$. There are two essential components of probability computation: (a) a density function $p_M$ and (b) a measure $dM$ over $M$. We assume $p_M$ and $dM$ to satisfy the followings.

- (**R1**) For any measurable subset $A \subset M$, i.e., $\Pr[\mathbf{x} \in A] = \int_{\mathbf{x} \in A} p_M(\mathbf{x}) dM(\mathbf{x})$.
- (**R2**) $p$ is zero everywhere out of $M$, i.e., $\text{supp}(p_M) = \{\mathbf{x} \in \mathbb{R}^n \mid p_M(\mathbf{x}) > 0\} \subseteq M$
- (**R3**) For any $(\mathbf{x}, y)$, $\mathbf{x}$ is sampled from $M_i$ if and only if $y = i$, i.e. $\Pr[\mathbf{x} \in M_i] = Pr[y = i]$

When equipped with such $p_M$ and $dM$, we call $M$ as a data-generating manifold.

**Probability over a Riemannian manifold.** We show how to compute a probability of $\mathbf{x}$ being generated from a Riemannian manifold $M$. We assume a $k$-dimensional manifold $M$ equipped with a Riemannian metric $g$, a family of inner products $g_{\mathbf{x}}$ on tangent spaces $T_{\mathbf{x}}M$. In this case, $g$ induces the volume measure $dM$ for integration over $M$. If $M$ is parameterized by $\mathbf{x} = X(\mathbf{u})$ for $\mathbf{u} \in D \subseteq \mathbb{R}^k$, the integration of a density function $p_M$ on $M$ is as follows.

$$\int_{\mathbf{x} \in M} p_M(\mathbf{x}) dM(\mathbf{x}) = \int_{\mathbf{u} \in D} p_M(X(\mathbf{u})) \sqrt{|\det[g_{X(\mathbf{u})}]|} d\mathbf{u} \tag{8}$$

where $[g_{X(\mathbf{u})}]$ is the $k \times k$ matrix representation of the inner product $g_{X(\mathbf{u})}$ at $X(\mathbf{u}) \in M$.

In Appendix B, a concrete example of this computation will be provided.

## D.2 DENSITY EXTENSION OF THE SECTION 3.2

This section introduces some remaining discussions regarding our data-generating process from a data-generating manifold.

**Relation to kernel density estimation.** While this extension is computing the density of compound distribution, it can be interpreted as computing expectation over a family of locally defined densities. Such an expected value can be observed in previous approaches of density estimation. For example, if $\nu_{\mathbf{x}}$ is isotropic Gaussian for each $\mathbf{x}$, the above integration is equivalent to the kernel density estimation, with Gaussian kernel, over infinitely many points on $M$.

**Observed property of the extended density.** In Figure 6b in Appendix B, we can observe that the extended density achieved higher values near the data-generating manifold. We formalize this observation to discuss its implication to the INC approach.

Let $d(\hat{\mathbf{x}}, M)$ to be the minimum distance from $\hat{\mathbf{x}}$ to the manifold $M$.

(**C1**) For any given $\hat{\mathbf{x}}$, let $y^*$ be the class label whose conditional density $p(\hat{\mathbf{x}}|y = y*)$ dominates $p(\hat{\mathbf{x}}|y = i)$ for $i \neq y^*$,

$$y^* \in \arg\max_{i \in [l]} p(\hat{\mathbf{x}}|y = i) \tag{9}$$

and let $M_{y^*}$ be the manifold corresponding to $y^*$.

(**C2**) For $y^*$ satisfying (**C1**), we choose $y^*$ such that the distance of $\hat{x}$ from the manifold $d(\hat{\mathbf{x}}, M_{y*})$ is the smallest.

If there are multiple $y^*$ satisfying both of (**C1**) and (**C2**), we expect the following property to be true for all of those $y^*$.

(**P1**) Consider the shortest line from $\hat{\mathbf{x}}$ to the manifold $M_{y^*}$. As $\hat{\mathbf{x}}$ goes closer to $M_{y*}$ along this line, $\hat{\mathbf{x}}$ should be more likely to be generated as the influence of noise decreases when moving away from the manifold. Therefore, we expect our density $p_M$ to have the following property.

$$\mathbf{x}^* \in \arg\min_{\mathbf{x} \in M_{y^*}} d(\hat{\mathbf{x}}, \mathbf{x})$$
$$\Longrightarrow p(\hat{\mathbf{x}}) \leq p((1 - \lambda)\hat{\mathbf{x}} + \lambda \mathbf{x}^*) \text{ for all } \lambda \in [0, 1] \tag{10}$$

Actually, this provides another justification of INC. In reality, the density conditioned by the label is not available even after running a generative model, so finding $y^*$ with (**C1**) is relatively hard. If we only consider (**C2**) without filtering $y^*$ via (**C1**), we are finding a point $\mathbf{x} \in M$ achieving the minimum distance to $\hat{\mathbf{x}}$, which is the optimization (11) above. Then projecting $\hat{\mathbf{x}}$ to the $\mathbf{x}^*$, i.e. the solution of the optimization 11, can be explained by 10; when $\lambda = 1$, $p$ is the highest along the shortest line between $\hat{\mathbf{x}}$ and $\mathbf{x}^*$.

## D.3 SUFFICIENT CONDITIONS FOR THE EXISTENCE OF RADII

We discuss the sufficient conditions guaranteeing the existence of radii introduced in Definition 1. Those sufficient conditions are derived from natural intuition about the properties of distributions in most machine-learning contexts.

The first intuition is that the influence of noise should diminish as observed sample $\hat{\mathbf{x}}$ moves away from a source point $\mathbf{x}_o$. Therefore, we formalize the noise whose density decreases as the noise $\mathbf{n} = \hat{\mathbf{x}} - \mathbf{x}_o$ gets bigger. We formalize boundedness of noise densities via the boundedness of their $\lambda$-density superlevel sets and continuity of noise density via the continuity of individual $\nu_{\mathbf{x}}$.

**Definition 11** (Center-peaked noise density)**.** Noise density functions $\nu_{\mathbf{x}}$ are *center-peaked*, if for any source point $\mathbf{x} \in M$ and any noise vector $\mathbf{n} \in \mathbb{R}^n$ with $\|\mathbf{n}\| > 0$, $\nu_{\mathbf{x}}(\mathbf{n}) < \nu_{\mathbf{x}}(\lambda \mathbf{n})$ for all $\lambda \in [0, 1)$.

**Definition 12** (Bounded noise density)**.** Noise density functions $\nu_{\mathbf{x}}$ are *bounded*, if a $\lambda$-density superlevel set is nonempty, there is a radius $\delta$ by which the $\lambda$-density superlevel set is bounded, i.e., $L_{\nu_{\mathbf{x}}, \lambda} \subseteq \overline{B_\delta(\mathbf{0})}$ where $\overline{B_\delta(\mathbf{0})}$ is the closed ball of radius $\delta$ centered at $\mathbf{0}$.

**Definition 13** (Continuous noise density)**.** Noise density functions $\nu_{\mathbf{x}}$ are *continuous*, if $\nu_{\mathbf{x}}$ is continuous in $\mathbb{R}^n$, for any $\mathbf{x} \in M$.

Under the conditions above, the radii in Definition 1 always exist.

**Proposition 1.** If noise densities $\nu_{\mathbf{x}}$ are center-peaked, bounded, and continuous, any nonempty $\lambda$-density superlevel set $L_{\nu_{\mathbf{x}},\lambda}$ has both $\lambda$-bounding radius $\delta_{\mathbf{x},\lambda}$ and $\lambda$-guaranteeing radius $\epsilon_{\mathbf{x},\lambda}$.

*Proof.* Let $\nu_{\mathbf{x}}$ be a center peaked, superlevel set bounded family of continuous noise densities. Since $\nu_{\mathbf{x}}$ is continuous, superlevel set $L_{\nu_{\mathbf{x}},\lambda} = \nu_{\mathbf{x}}^{-1}[\lambda, \infty)$ is closed as an inverse image of $\nu_{\mathbf{x}}$. Therefore, its boundary $\partial L_{\nu_{\mathbf{x}},\lambda}$ is contained in $L_{\nu_{\mathbf{x}},\lambda}$.

Because $\nu_{\mathbf{x}}$ is superlevel set bounded, superlevel set $L_{\nu_{\mathbf{x}},\lambda}$ is bounded by a closed ball $\overline{B_\delta(\mathbf{0})}$ with radius $\delta \geq 0$. Since $\nu_{\mathbf{x}}$ is center peaked, a nonempty superlevel set $L_{\nu_{\mathbf{x}},\lambda}$ always contains $\mathbf{0}$ as the maximum is achieved at $\mathbf{0}$. Moreover, there exists a closed neighborhood ball $\overline{B_\epsilon(\mathbf{0})}$ with radius $\epsilon \geq 0$ contained in the superlevel set $L_{\nu_{\mathbf{x}},\lambda}$. Now it is enough to show that the minimum of $\delta$ and the maximum of $\epsilon$ exist.

Since $L_{\nu_{\mathbf{x}},\lambda}$ is bounded, its boundary $\partial L_{\nu_{\mathbf{x}},\lambda}$ is also bounded. $\partial L_{\nu_{\mathbf{x}},\lambda}$ is closed and bounded, thus it is a compact set. Therefore, the Euclidean norm, as a continuous function, should achieve the maximum $\overline{r}$ and the minimum $\underline{r}$ on $\partial L_{\nu_{\mathbf{x}},\lambda}$ by the extreme value theorem. From the choice of $\delta$ and $\epsilon$, we can get,

$$\epsilon \leq \underline{r} \leq \overline{r} \leq \delta$$

Therefore, we can find the minimum $\delta_{\mathbf{x},\lambda} = \overline{r}$ and the maximum $\epsilon_{\mathbf{x},\lambda} = \underline{r}$. $\qquad\square$

### D.4 GENERALIZATION OF THE THEOREM 1

We try generalizing the Theorem 2 to handle more concepts in topology. Theorem 2 mainly uses a fact that the number of connected components of $\lambda$-density superlevel set is preserved by a continuous generative model $G$.

In algebraic topology, each connected component corresponds to a generator of 0-th homology group $H_0$, and continuity of a function is enough to preserve each component. In general, generators of $i$-th homology group $H_i$ for $i > 0$ are not preserved by a continuous map, so we need to restrict $G$ further. By requiring $G$ to be a homeomorphism, we can safely guarantee that all topological properties are preserved by $G$; therefore, we can generalize the Theorem 2 with a homeomorphic generative model $G$.

To generalize the proof of the Theorem 2, we first provide the sketch of the proof.

(1) $\lambda^*$-density superlevel set $L_{\lambda^*}^Z$ of a mixture of $n_Z$ Gaussian distributions has at most $n_Z$ connected components.
(2) Since $G$ is continuous, the number of connected components of $L_{\lambda^*}^G(Z) = G(L_{\lambda^*}^Z)$ is same to the number of connected components of $L_{\lambda^*}^Z$, so it is also at most $n_Z$.
(3) We choose $\lambda^*$ so that $L_{\lambda^*}^X$ is included in $\delta^*$-neighborhood of $M$.
(4) By assumption on the class-wise distance of $M$, $\delta^*$-neighborhood of $M$ has exactly same number of connected components to $M$, i.e., $n_X$. Therefore $L_{\lambda^*}^X$ has at least $n_X$ connected components.
(5) By (2) and (4), we conclude that $L_{\lambda^*}^G(Z)$ and $L_{\lambda^*}^X$ do not agree on the number of connected components as long as $n_Z < n_X$.

In this proof, $n_Z$ corresponds to the maximal 0-th Betti number of $L_{\lambda^*}^Z$, i.e. the number of generators of $H_0(L_{\lambda^*}^Z)$. If we keep using a mixture of Gaussians as latent vector distribution, all components of $L_{\lambda^*}^Z$ are contractible, so we may use 0 as the maximal $i$-th Betti number.

Also, $n_X$ corresponds to the 0-th Betti number of $M$ and it worked as the minimal 0-th Betti number of $L_{\lambda^*}^X$. The condition on the class-wise distance of $M$ is used to ensure $n_X$ to be a lower bound. Combining these observations, we can get the following generalized statement.

**Theorem 3.** Let $\mathcal{D}_Z$ be a mixture of multivariate Gaussian distributions, and let $\mathcal{D}_X$ be the target distribution from data-generating manifold $M$. Let $n_i$ be the $i$-th Betti number of $M$.

Consider a generative model $G$ is used to approximate $\mathcal{D}_X$ using the latent vectors sampled from $\mathcal{D}_Z$. Assume that $G$ is a homeomorphism from $\mathbb{R}^n$ to itself. Assume that data-generating manifold satisfies the conditions of the Theorem 1, and let $\lambda^*$ be the threshold value that $L_{\lambda^*}^X$ corresponds

to that superlevel set. Assume that for some $j > 0$, the homomorphism $\iota^*$ induced by the inclusion $\iota : M \to N_{\delta^*}(M)$ is injective. [5]

If $0 < n_j$, $L_{\lambda^*}^X$ and $L_{\lambda^*}^{G(Z)}$ do not agree on the number of connected component.

*Proof.* Since $L_{\lambda^*}^X$ is the results of Theorem 1, it includes $M$ and is included by $\delta^*$-neighborhood $N_{\delta^*}(M)$ of $M$. Define inclusions $\iota_1, \iota_2$ as,

- $\iota_1 : M \to L_{\lambda^*}^X$
- $\iota_2 : L_{\lambda^*}^X \to N_{\delta^*}(M)$

Clearly, $\iota = \iota_2 \circ \iota_1$.

Let $\iota_1^*$ and $\iota_2^*$ be induced homomorphisms of $\iota_1$ and $\iota_2$, resp.

By the assumption, any generator $[a]$ in $H_j(M)$ is injectively mapped to a nonzero generator $\iota^*([a])$ in $H_j(N_{\delta^*}(M))$. Therefore, the $j$-th Betti number of $N_{\delta^*}(M)$ is equal to that of $M$, i.e. $n_j$. Note that $j$-th Betti number is the rank of $j$-th homology group $\text{rank}(H_j(N_{\delta^*}(M)))$ Because $\iota_2^*$ is a homomorphism from $H_j(L_{\lambda^*}^X)$ to $H_j(N_{\delta^*}(M))$, $\text{rank}(L_{\lambda^*}^X) \geq \text{rank}(H_j(N_{\delta^*}(M)))$. Therefore the $j$-th Betti number of $L_{\lambda^*}^X$ is at least $n_j$.

However, since $\mathcal{D}_Z$ is a mixture of Gaussians, for any value of $\lambda$ (including the special case $\lambda = \lambda^*$), $L_\lambda^Z$ does not have any generator of $j$-th homology group, so it has $j$-th Betti number $0$ for all $j > 0$. Since $G$ is homeomorphic, it preserves all the Betti numbers, thus $L_{\lambda^*}^{G(Z)} = G(L_{\lambda^*}^Z)$ has the same $j$-th Betti number. As $0 < n_j$, $L_{\lambda^*}^X$ and $L_{\lambda^*}^{G(Z)}$ can never agree on the number of connected components. $\square$

In Section 5.2, we see the Figure 3i from the circles dataset, which is a remarkable example that $L_\lambda^G(Z)$ has the same number of connected components, but does not have any loop (non-contractible circle). This is empirical evidence of Theorem 3, so it is explained by mismatches in the topology of distributions. Each concentric circle has $\mathbb{Z}$ as its first homology group as circle contains exactly one generator. However, latent vector distribution always has a trivial first homology group, as any superlevel set of a mixture of Gaussians is a set of contractible connected components.

## D.5 DETAILS OF INC IMPLEMENTATIONS IN THE SECTION 5

**INC implementation.**  We start from introducing the optimization for the ideal INC projection when the data-generating manifold $M$ is available.

$$\mathbf{x}^* = \arg\min_{\mathbf{x} \in M} d(\mathbf{x}, \hat{\mathbf{x}}) \tag{11}$$

where $d$ is a metric defined on the domain $X$. If perfect classification on $M$ is assumed (model is well-trained on $M$) and $\hat{\mathbf{x}}$ is close enough to the manifold of correct label, classification $f(\mathbf{x}^*)$ is likely to be correct, since $\mathbf{x}^*$ is likely to lie on the correct manifold. Since the data-generating manifold $M$ is unknown, the INC approach runs the following optimization with before the classification.

$$\mathbf{x}^* = G(\mathbf{z}^*) \text{ where } \mathbf{z}^* = \arg\min_{\mathbf{z} \sim \mathcal{D}_Z} d(G(\mathbf{z}), \hat{\mathbf{x}}) \tag{12}$$

where $d$ is a metric defined on the domain $X$.

When INC is implemented with a reversible generative model $G$, for any given $\hat{\mathbf{x}} \in \mathbb{R}^n$ there exists a trivial solution $\mathbf{z}^* = G^{-1}(\hat{\mathbf{x}})$ to the optimization (12), achieving $d(G(\mathbf{z}^*), \mathbf{x}) = 0$. This is even true for $\hat{\mathbf{x}}$ out of the manifold, resulting in the situation that the output $\mathbf{x}^* = G(\mathbf{z}^*) = \hat{\mathbf{x}}$ is still out of the data-generating manifold.

To manage this problem, we add another term penalizing a low density of latent vector to the objective function. Thus, in our INC implementation, we solve the following optimization problem.

$$\mathbf{x}^* = G(\mathbf{z}^*) \text{ where } \mathbf{z}^* = \arg\min_{\mathbf{z} \sim \mathcal{D}_Z} [d(G(\mathbf{z}), \hat{\mathbf{x}}) + \alpha(M - p_Z(\mathbf{z}))] \tag{13}$$

---

[5]Any generator of the $j$-th homology group $H_j(M)$ of $M$ is mapped to a nonzero generators of the $j$-th homology group $H_j(N_{\delta^*}(M))$ of $\delta^*$-neighborhood of $M$.

where $\alpha$ is the regularization factor and $M$ is the maximum possible value of the density $p_Z$ of the latent vector distribution. For the choice of regularization factor, we used the same value $\alpha = 1$ during the entire experiment.

To solve each optimization problem, we used the built-in adam optimizer (Kingma & Ba, 2014) in Tensorflow package. For optimization parameters, we ran 100 iterations of adam optimizer using learning rate 0.01 with random sampling of $\mathbf{z}$.

When implementing INC using a class-aware generative model, we used the following strategy to improve its robustness.

- As the class-aware generative model generates each manifold from each Gaussian component, we first sample initial points from each manifold by randomly choosing latent vectors $\mathbf{z}_1, \ldots, \mathbf{z}_l$ from each Gaussian component.
- We run INC for $i$-th manifold by solving the following optimization.

$$\mathbf{x}_i^* = G(\mathbf{z}_i^*) \text{ where } \mathbf{z}_i^* = \arg\min_{\mathbf{z} \sim \mathcal{D}_Z} \left[ d(G(\mathbf{z}), \hat{\mathbf{x}}) + \alpha(M_i - p_{Z,i}(\mathbf{z})) \right]$$

where $M_i$ is the maximum value of $i$-th Gaussian component. The regularization term is designed to penalize $\mathbf{z}$ which is unlikely to be generated by $i$-th Gaussian component, so we only search in the range of $i$-th Gaussian component, i.e., $i$-th manifold.
- We choose the final solution $\mathbf{x}_i^*$ achieving the minimum $d(\mathbf{x}_i^*, \hat{\mathbf{x}})$, breaking ties randomly.

Since each search is performed only on each submanifold, the artifact observed in Section 5.3 never appears during the optimization process. Also, choosing initial points from each manifold prevents the initialization problem mentioned in Section 5.3.

### D.6 DISCUSSION ABOUT THE LIMITATION OF TOPOLOGICAL INFORMATION

Given a sufficient number of connected components in the latent vector distribution, does the class-aware training suggested in this paper result in a generative model that achieves manifold separation? For this question, the answer is **no**, and the manifold separation depends on other factors, e.g., alignment of latent vector distribution, choice of training parameter, etc.

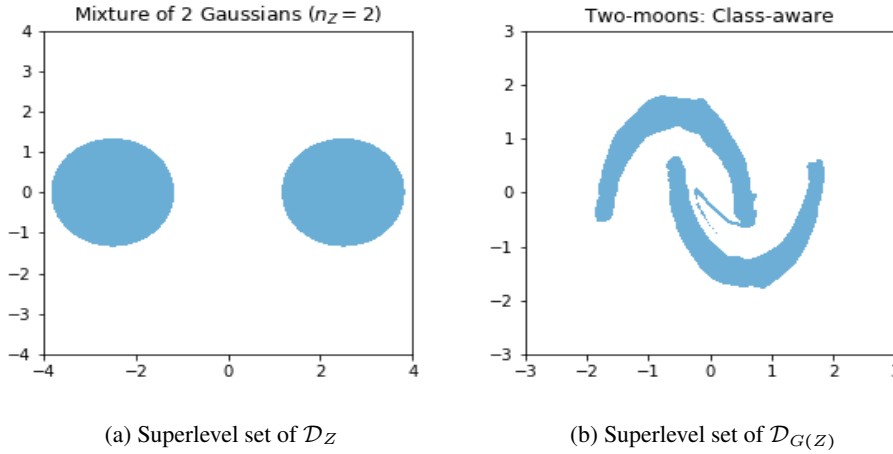

(a) Superlevel set of $\mathcal{D}_Z$  (b) Superlevel set of $\mathcal{D}_{G(Z)}$

Figure 7: Failure cases of class-aware training.

Figure 7b shows the superlevel set of $\mathcal{D}_{G(Z)}$ from a class-aware training to learn the two-moons dataset when latent vector distribution is a mixture of two Gaussian distributions aligned horizontally (Figure 7a). It is clear that in this case, the generative model induced a connection artifact even when the class-aware training was used.

We explain this by interpreting reversible generative models as dynamical systems (Weinan, 2017; Chen et al., 2018; Grathwohl et al., 2018; Zhang et al., 2018). To elaborate, a reversible generative

model can be viewed as a dynamical system moving the latent vector distribution to the target distribution continuously in time. When two Gaussian mixtures are aligned vertically, a reversible generative model is likely to learn how to move the upper (and lower) Gaussian distribution toward the upper moon (and the lower moon, respectively), without being affected by the entanglement of two moons. However, moving the left (and right) Gaussian distribution toward the left moon (and the right moon, respectively) continuously in time is required to avoid the entanglement of two moons during the transition. This case alludes that information about the topological properties may not be enough to learn a generative model separating manifolds, because it does not provide an understanding of information about how data-generating manifolds are aligned.

## E    MORE EXPERIMENTAL RESULTS

We present more experimental results about the INC performance comparing topology-aware generative model to its topology-ignorant counterpart.

**Histogram for projection error distributions in 5.4.**    Figure 8 presents the histogram of the projection errors distributed from 0 to the diameter of the distribution. Each row corresponds to each dataset, whereas the first column and the second column represent the results from the topology-ignorant model and the topology-aware model, respectively. All histograms are normalized so that the sum of values adds up to 1. To explain, the $y$-axis of each histogram is the estimated probability that INC achieves the projection error on the $x$-axis. Not only can we observe the improved mean of projection errors in the histograms, but we can also check the reduced standard deviation, i.e., we get more consistent projection errors near the mean.

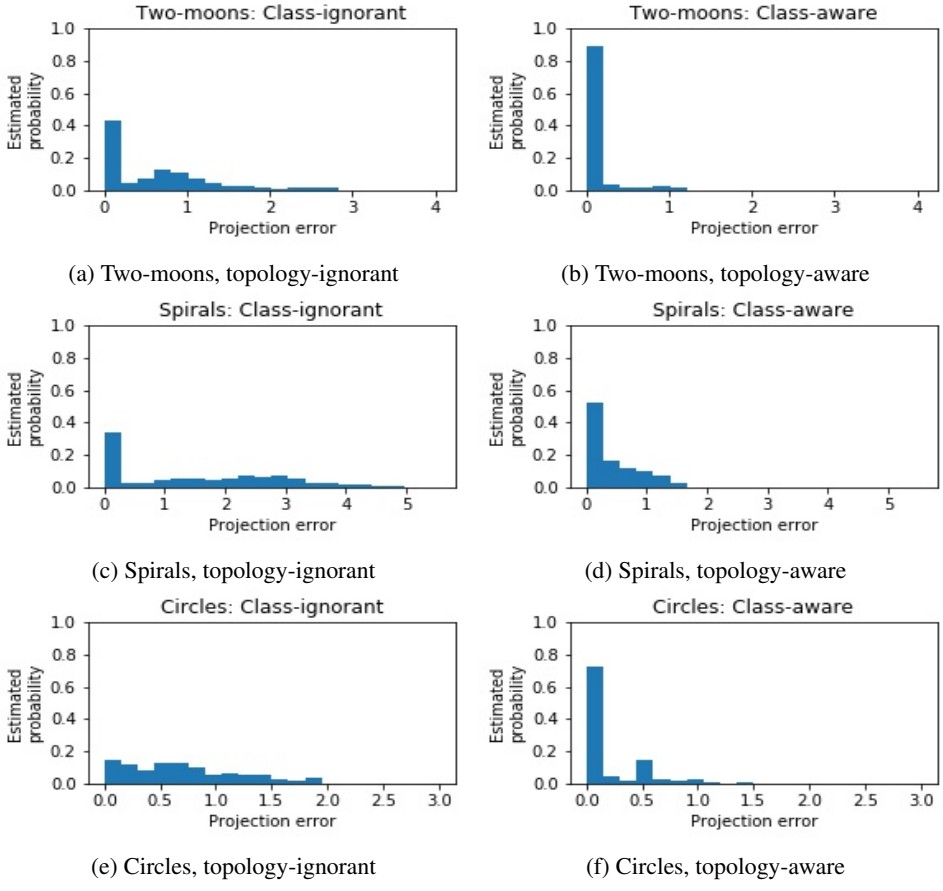

(a) Two-moons, topology-ignorant

(b) Two-moons, topology-aware

(c) Spirals, topology-ignorant

(d) Spirals, topology-aware

(e) Circles, topology-ignorant

(f) Circles, topology-aware

Figure 8: Histograms of the projection errors of INC. Each $y$-axis represents the estimated probability that INC incurs the projection error on the corresponding $x$-axis.

