# OpenReview forum: "On the Need for Topology-Aware Generative Models for Manifold-Based Defenses"
_ICLR.cc/2020/Conference — Accept (Poster)_

### Official Review · AnonReviewer1 · 2019-10-20
**Official Blind Review #1**

**Rating:** 6

**Review:**

I. Summary of the Paper

This paper studies robustness to adversarial examples from the
perspective of having 'topology-aware' generative models. Next
to some experiments on data sets with a manifold structure, the
main contribution of the paper is a tandem of theorems that state
the conditions under which models can recover the topology---or
the number of connected components---of a data set correctly,
thereby making them more robust to adversarial examples.

II. Summary of the Review

This paper provides a novel perspective on adversarial examples through
the lens of Riemannian Geometry and topology. I appreciate novel
research that employs topology-based methods, but at present, I cannot
fully endorse accepting the paper. Specifically, I see the following
issues:

- Missing clarity: while the appendix is very comprehensive, which
  I appreciate, the main text could be improved; some statements appear
  redundant, while others need to be re-formulated to build intuition

- This paper appears to span both theory and applications. I appreciate
  this attempt, knowing full well that this is no easy feat to
  accomplish. However, the main theoretical result on the number of
  connected components only applies to mixtures of Gaussian
  distributions, but the purported scope of the paper is the analysis of
  manifold-based defences in general. I would expect a more in-depth
  discussion of the limitations of the theorem. Can we expect this to
  generalise? Moreover, 'topology' is reduced to 'connected components'
  in this paper. While this is perfectly adequate in the sense of
  connected components being a particular concept from topology, I would
  expect this to be clarified much earlier in the paper. In addition,
  connected components are a very basic and coarse concept, so I am
  wondering to what extent it is sufficient to describe models purely
  based on that information.

- As a sort of corollary to the previous point, the experiments could be
  improved. I like the idea of employing known data sets with a simple
  manifold structure, but the setup is somewhat preliminary; I would
  prefer to see an analysis of border cases or limit cases in which the
  theorem _almost_ applies (or not); plus, a more in-depth analysis of
  stochastic effects during training: do _all_ models end up being
  robust if their number of connected components is sufficiently large?
  Is there a dependency between the number of connected components and
  vulnerabilities---are models with a very small number of connected
  components more vulnerable than models with a very larger number of
  connected components? The present experimental section is lacking this
  depth.

- The same statements apply to the INC example. I found this super
  instructive, but it is only _one_ case on _one_ manifold---I would
  like to see more details here; maybe some of the experiments in the
  appendix could be moved to the front? I have some suggestions for
  shortening the paper (see below).

Despite these issues, I think this paper can be a strong contribution if
properly revised; since I am positive that at least some of these
suggestions could be performed within a revision cycle, I want to be
upfront and state that I will definitely consider raising my score,
provided that my concerns are addressed appropriately!

I have to state that I am _not_ an expert in adversarial examples, but
an expert in topology-based methods; I consider this paper to belong to
the latter field given its theoretical contributions about 'recovering'
the correct density distribution.

III. Clarity

The paper provides an extensive background to Riemannian geometry, which
I appreciated as a reference. Nevertheless, there are improvements to
the main text that I would suggest:

- Please consider changing the title to 'On the need...'

- The manifold assumption is that data lie _on_ a manifold or _close to_
  a manifold whose intrinsic dimension is much lower than that of the
  ambient space. This is not stated in sufficient precision in the
  paper; please correct the usage on p. 1 and p. 2

- In terms of notation, why use $p_M$ to denote the density on the whole
  of $\mathds{R}^n$? I would expect $p_M$ to refer to the density on $M$
  rather than the density of the whole space.

- If $M$ is a disjoint union of manifolds, please consider using
  a '\cupdot' operator to make this more clear.

- If the pairwise manifolds are disjoint, how can the resulting data
  distribution still contain any ambiguities? I find this hard to
  harmonise with the statement in Section 3.4 about the existence of
  a classifier that separates the manifolds. Please clarify the meaning
  behind the term 'ambiguities' here.

- The '(R0)' requirement definition and the discussion in Section 3.2
  strike me as needlessly complex. Would it be possible to shorten this
  or move some content to the appendix? I think it would be sufficient
  to have Eq. 1 and mention how it could be solved.

- Section 3.3 could be shortened as well, if I am not mistaken; while it
  is good to know how such models look, the 'change of variable formula'
  is not used directly any more in the paper; I think it might be easier
  to write down a generic form of the density for each model.

- The salient points of Section 3.4 seem to be the projection point;
  maybe this could also be shortened somewhat in the interest of having
  more space for experiments. The relevant information of this section
  was to learn how projections work for different models, but it would
  be sufficient to keep Eq. 3 and discuss Eq. 4 in the appendix

- The 'topological differences' mentioned in Section 4 could be
  clarified: the paper talks about differences in connected components.

- The $\lambda$-density set appears to be a superlevel set, if I am not
  mistaken: the level set would be defined for a single threshold only,
  while this paper introduces the pre-image of an interval.

- I would expect the pre-image to be defined as $p^{-1}$, not $p_{-1}$;
  the latter strikes me as somewhat non-standard usage

- The statements preceding Definition 1 require some more intuition;
  what is the purpose of these assumptions?

- The notation for the Euclidean balls should be briefly introduced
  before Definition 1. I recognise this as a standard notation, but
  since this is the first appearance of the symbol, it should be
  mentioned at least briefly.

- Definition 1 could also be phrased more intuitively; additional
  sentence behind each definition would be useful, such as:
  $\delta_\lambda$ is the largest $\delta$ such that the full
  superlevel set is contained in a ball of radius $\delta_\lambda$.

  Figure 1 is already helpful in that regard; it should ideally precede
  the definition and/or be made larger to be more illustrative

- Maybe the results of Theorem 2 could already be stated earlier; it
  could probably be explained reasonably well when describing the number
  of connected components as a sort of 'baseline' topological complexity
  that a model has to satisfy.

  On a more abstract level, could it also be summarised as 'the
  inclusion of prior knowledge is a necessary condition for robustness'?

- I would not state that the main goals of the experiments are to
  'check the correctness of Theorem 2'---the proofs should be
  responsible for this! I would rather say that the main goals are to
  provide empirical evidence for the *relevance* or *applicability* of
  the Theorem.

- The paragraph on 'Latent vector distributions' contains the most
  relevant information, viz. the knowledge of what the paper considers
  to be a 'topology-aware' and 'topology-ignorant' model; this should be
  highlighted more; maybe the figures could be extended to contain
  information about $n_X$?

Overall, I like the idea of having one overarching question in a paper
that is subsequently answered or discussed under different aspects. I
very much commend the authors of the paper for choosing this sort of
writing style!

IV. Experimental setup

As mentioned above, the experiments require more depth. I would propose
adding more repetitions of the training process for different data sets
and analysing the impact of the 'parameters' used in the theorems.

In particular, the 'INC' experiments show great promise for multiple
repetitions. Why not choose more data sets and more staring positions
and visualise the trajectories of _multiple_ draws, as shown for
a single draw in Figure 4 a,b,c?

Also, please consider moving the additional experiments from the
appendix to the main paper.

More compelling examples would also be helpful. Why not generate data
sets consisting of more than two manifolds? At present, the largest
issue I see in this section is that the conceptual 'leap of faith'
between the theory and the applications is simply too large. Would it
not be possible to perform the same experiments on a simple digits data
set, say MNIST?

V. Minor issues

The paper is well-written overall. There are only a few typos and minor
style issues that I would recommend fixing:

- please check the usage of quotes; it should be ``pulled back'', not ''pulled back'' in LaTeX

- please check the usage of citations; if '\citet' is used, citations
  can be used as nouns directly (for example: 'in Pennec (1999)' instead
  of 'in (Pennec, 1999)'.

- etc.. --> etc.
- it approximates posterior distribution --> it approximates a posterior distribution
- for compound distribution --> for a compound distribution
- a relatively simpler distribution --> a simpler distribution
- near manifold --> near a/the manifold
- we simply the minimum --> we denote (?) the minimum
- satisfies the followings --> satisfies the following properties
- number of connected component --> number of connected components
- Experimental result --> Experimental results

VI. Update after the rebuttal

The authors managed to address the majority of my comments. Overall, I still would like to see a more detailed/in-depth experimental setup, but I realise that this not directly possible within the timeframe allotted during the rebuttal period. I am thus raising my score.

**Experience Assessment:**

I have published in this field for several years.

**Review Assessment: Checking Correctness Of Derivations And Theory:**

I carefully checked the derivations and theory.

**Review Assessment: Checking Correctness Of Experiments:**

I carefully checked the experiments.

**Review Assessment: Thoroughness In Paper Reading:**

I read the paper thoroughly.

---

> ### Author Response · Authors · 2019-11-15
> **Thank you for your valuable feedback!**
>
> We highly appreciate your detailed feedback and all your suggestions.
>
> **Issues pointed out in the summary**
> We see that the issues mentioned in this section are truly valuable comments as they inspire lots of possible future works.
>  - For the issue about the clarity, we followed the suggestion items appears in “III. Clarity”
>  - While Gaussian distributions are very common choices for latent vector distribution, we agree that the theorem can be generalized to a more broad family of distributions. We consider this generalization as the main theme of our future work.
>
> **Clarity**
> Thank you for all of the thoughtful suggestions to improve the clarity of the work.
> We checked the items one by one and applied the following changes for each item.
>  - We first changed the title to "On the Need For Topology-Aware Generative Models for Manifold-based Defenses".
>  - We clarified the manifold assumption (in p. 2) as suggested in the comment.
>  - We exchanged all p and p_M as p_M is a more proper notation for density defined only on M.
>  - "\cupdot" operator is now used to denote a disjoint union of manifolds.
>  - The statement containing the term 'ambiguities' was removed as it is misleading and redundant.
>  - The requirement (R0) was moved to the appendix.
>  - "Change of variable formula" was moved to the appendix once, but now in section 5.1 to describe how training loss is estimated.
>  - Both "INC optimization formulae" (eq.3 & eq.4 in the previous version) are moved to the appendix.
>  - In the first paragraph of section 4, now we clarify that we specifically discuss the number of connected components.
>  - The term “level set” is replaced by “superlevel set”.
>  - The preimage notation $p_{-1}$ was a typo, so corrected to $p^{-1}.
>  - We introduced the notation for the Euclidean ball, before the notation is used.
>  - Additional sentences were added to describe each radius.
>  - Figure 1 was moved to the position near the Definition 1.
>  - We agree that the correctness of Theorem 2 is clear from the proof. We now express the goal of the 1st experiment as “empirical support for the applicability of Theorem 2”
>  - We extended the figure 3 to show the superlevel set components of the latent vector distribution.
>
> **Experimental setup**
> We moved the additional experiments from the appendix to the main paper, as a visual comparison of decision boundaries can be illustrative to show the effect of our training method.
>
> We prioritized the experiment on high dimensional data (e.g. MNIST), to show the applicability of our method in a more practical domain. We spent the majority of the revision period conducting experiments for the MNIST dataset. However, we are unable to obtain convincing results in the time frame provided.
>
> We used our own code base to implement the realNVP generative model for MNIST dataset for the following reasons.
> We wanted to re-use the code for INC projection without dealing with technical difficulties stemming from using other implementations.
> There is no realNVP implementation (for MNIST) that standardizes data for preprocessing. This standardization is useful for us in choosing the centers of Gaussian distributions (when constructing the mixture of Gaussian for latent vector distribution) for class-aware training, as the dataset is scaled adaptively by its own standard deviation.
> However, after all the experiments, it turned out that the generative model we trained is not good enough to be used (in terms of the quality of images that it generates) for INC defense.
>
> Training a generative model for high dimensional data is a challenging topic by itself, and we lack experience in this field. We will place top priority on implementing our idea for higher dimensional data, to support the wide applicability of our work.
>
> Due to the major time loss made in the MNIST experiment, we are not able to make all the other useful suggestions on experiments. However, for the question “Do all models end up being robust if their number of connected components is sufficiently large?”, we already knew that the answer is “no” in general. Training a generative model is influenced by choices of other hyperparameters, and we added one section describing the case that “the alignment of Gaussian matters”. We also provided a hypothetical explanation (of the reason) by interpreting the reversible generative model as a dynamical system moving the latent vector distribution to the target distribution.
>
> **Minor issues**
> We appreciate your kindness in telling us about those typos and style issues. Now, all issues mentioned are corrected.

---

> > ### Comment · AnonReviewer1 · 2019-11-15
> > **Thanks for the rebuttal**
> >
> > Thank you very much for the detailed rebuttal. I very much appreciate that you tried to incorporate so many of my comments/suggestions, and I fully understand that the integration of yet another data is not easy to do. I am really happy to see that the paper gained a lot of clarity and depth, and I am thus raising my score.
> >
> > I am not sure whether this is still possible within a short timeframe, but would you be able to briefly comment on what issues you encountered with MNIST? I would be really interested in the progress that you made, even though I can also understand that the time was insufficient to add such a suite of experiments to the paper.

---

> > > ### Author Response · Authors · 2019-11-15
> > > **Thanks for your response**
> > >
> > > Thank you for another response.
> > > The following challenges that we encountered come across our mind.
> > >
> > >  - Our code was originally designed for 2-D data, and it required a few more implementations for high dimensional data, so there have been major changes in the code to incorporate the change in the architecture. For example, we implemented checkerboard masking (by permuting dimensions before passing data into coupling layers) as it is not explicitly supported by the TensorFlow-probability library.
> > >
> > >  - As the loss function of training is defined with negative log-likelihood, when the model encounters a training batch that is very unlikely to be generated by the model being trained, the loss blows up to infinity value (np.inf in Python Numpy library) and fails to compute the gradient for next training. This happened from time to time, crashing the training during the convergence. After numerous trials and errors, we found a parameter setting that reduces the number of these mishaps, so now the training process at least decreases the model loss.
> > >
> > >  - Applying class-aware training to real-world data, e.g. MNIST, was another challenge. For example, from the new discussion in Appendix D.6, we know that the alignment of Gaussian distribution matters, and it is desirable to find the center of Gaussians that,
> > >   1. Reflects the “location” of each data-generating manifold corresponding to each label.
> > >   2. Each center should be far enough so that superlevel sets of Gaussian are separated in the latent vector space.
> > > We first computed the center of each cluster (data points with the same label) in the training set, to reflect the location of each data-generating manifold, then we took the elementary basis of $R^784$ (784 is the dimension of MNIST dataset) that is closest to the center. By taking elementary basis elements we can ensure the minimum distance is bounded by some number so that superlevel sets of Gaussian are separated with a proper choice of the standard deviation of Gaussian distributions.

---

### Official Review · AnonReviewer2 · 2019-10-24
**Official Blind Review #2**

**Rating:** 8

**Review:**

This paper argues that defenses against adversarial attacks need to be stronger than they currently are. Defenses that use generative models assume that there exists a manifold of data that is modeled by a trained generative model that can be used to project any out-of-manifold data unto the manifold. However, this work argues that if the generative model does not model the topology of the manifold, it can still be fooled by an adversarial example. They argue that a generative model needs to be at least aware of the number of connected components of the data-generating manifold. If the number of connected components does not match, based on theorem 2, Corollary 1 argues that a generative model can generate an adversarial example that does not exist in the data-generating manifold.

Pros:
- To the extent I checked, proofs are correct.
- The experimental results support this result on 2D toy manifolds. They show how a prior defense based on generative-models (INC) fails on the toy problems and show how a modification to INC can improve it.

Cons:
- In their experiments, they use the number of classes as an approximation to the number of connected components (Appendix E) and train class-conditional generative models. Some of these details are better to be put in the main text.
- There are no experiments beyond toy examples on high-dimensional problems and datasets. It should not be too difficult to have some preliminary results using the proposed extension of INC on MNIST or CIFAR10.

After rebuttal:
I have raised my score after authors improved to quality of the text. Even though this work does not have empirical results on high-dimensional datasets such as MNIST or CIFAR10, it has a nice theoretical contribution useful for finding stronger defenses against adversarial examples.

**Experience Assessment:**

I have published one or two papers in this area.

**Review Assessment: Checking Correctness Of Derivations And Theory:**

I assessed the sensibility of the derivations and theory.

**Review Assessment: Checking Correctness Of Experiments:**

I assessed the sensibility of the experiments.

**Review Assessment: Thoroughness In Paper Reading:**

I read the paper at least twice and used my best judgement in assessing the paper.

---

> ### Author Response · Authors · 2019-11-15
> **Thank you for your valuable feedback!**
>
> We appreciate your review and all the suggestions.
>
> **Notational Changes**
> In case you read the revised version, we made the following notational changes for the clarity of the paper.
>  - We exchanged all p and p_M, as p_M is a more proper notation for density defined only on M.
>  - "\cupdot" operator is now used to denote a disjoint union of manifolds.
>  - The term “level set” is corrected to a more proper term “superlevel set”.
>  - The preimage notation $p_{-1}$ in previous version was a typo, so we corrected it to $p^{-1}.
>
> **Major changes**
>  - We mostly re-structured the paper, so that we can move details about training class-conditional generative model and an additional experiments.
>  - Appendix D.6 was added to discuss the limitation of topological information, suggesting that more detailed information about data-generating manifold may be exploited to train a better model.
>
> **Changes reflecting the comments**
>  - We re-structured the paper based on your feedback. Specifically, we have moved details regarding training of the class-conditional generative models to the main text. Your feedback also suggested we discuss in detail how one could implement INC for class-conditional generative models. However, we decided to leave this discussion in the appendix to create space for experiments that required to be moved from the appendix to the body of the paper.
>
> Your feedback suggested we expand our experiments are to include results from high dimensional models such as MNIST. We spent the majority of the revision period conducting experiments for the same. However, we are unable to obtain convincing results in the time frame provided.
>
> We used our own code base to implement the realNVP generative model for MNIST dataset for the following reasons.
> We wanted to re-use the code for INC projection without dealing with technical difficulties stemming from using other implementations.
> There is no realNVP implementation (for MNIST) that standardizes data for preprocessing. This standardization is useful for us in choosing the centers of Gaussian distributions (when constructing the mixture of Gaussian for latent vector distribution) for class-aware training, as the dataset is scaled adaptively by its own standard deviation.
> However, after all the experiments, it turned out that the generative model we trained is not good enough to be used (in terms of the quality of images that it generates) for INC defense.
>
> Training a generative model for high dimensional data is a challenging topic by itself, and we lack experience in this field. We will place top priority on implementing our idea for higher dimensional data, to support the wide applicability of our work.

---

### Official Review · AnonReviewer3 · 2019-11-01
**Official Blind Review #3**

**Rating:** 3

**Review:**

The paper tries to answer the following question:
In adversarial defense training do manifold based defenses need to know the structure of the underlying data manifold?
The question is quite rhetoric (the answer is most probably yes), nevertheless, the paper provides a theoretical and empirical answer.

The paper reads well and it is interesting to read. Nevertheless, I have a very basic question regarding the usefulness of the methods that the paper studies and the topic of adversarial defenses. I have worked on the topic for some years and in the beginning, I found it quite interesting, until I realized that, at least for images, audio, 3d meshes, all adversarial attacks can be very easily addressed with simple denoising mechanisms (for images even a non-local means filter or even a Gaussian filter eliminated all the adversarial attacks I have tried). There are some recent papers that demonstrate this [A] or recently feature denoising [B]. Why denoising is not enough to pull the data back to the data manifold (e.g., general low-rank data denoising or general denoising suitable for the data under investigation)?

I really want a discussion about that before I make a final decision.

[A] Defensive denoising methods against adversarial attack
[B] Feature Denoising for Improving Adversarial Robustness

**Experience Assessment:**

I have read many papers in this area.

**Review Assessment: Checking Correctness Of Derivations And Theory:**

I assessed the sensibility of the derivations and theory.

**Review Assessment: Checking Correctness Of Experiments:**

I assessed the sensibility of the experiments.

**Review Assessment: Thoroughness In Paper Reading:**

I read the paper at least twice and used my best judgement in assessing the paper.

---

> ### Author Response · Authors · 2019-11-14
> **Thank you for your valuable feedback!**
>
> Thank you for your time and effort spent on the feedback.
> Also, thank you for mentioning two related work regarding defense strategies based on denoising adversarial effects. We will add a discussion to these papers in our related work section.
>
> We want to reiterate that the main theme of our work is "The need of topology-awareness of a generative model when a generative model is exploited as a part of defense" rather than "The effectiveness of denoising as a defense mechanism". Both [A] and [B] don't seem to have applications of a generative model, but we enjoyed reading both [A] and [B].
>
> For the question "Why denoising is not enough to pull the data back to the data manifold?", we believe that denoising should be an effective way to implement the projection to manifold. Assuming that denoising is done with a proper choice of parameters, it ideally removes the noise and pulls the data back to the data generating manifold. However, before the work of [C], there have been trials (e.g. [D], [E], [F]) and errors (e.g. [G], [H], [C]), regarding denoising images in pixel space, that cast doubt on the research question of "Effectiveness of denoising as a defense mechanism".
>
> Focusing on the topic of "Effectiveness of denoising as a defense mechanism", there are more themes to discuss. Before further discussions on both of [A] and [B], we summarize a related work [C]. In [C], an intensive case study was performed to test their attack that circumvents obfuscated gradients that are the basis of most defense mechanisms. Especially, they introduced Backward Pass Differentiable Approximation (BPDA) to circumvent the part, of the defense mechanism, that obfuscates gradients by replacing the part by differentiable approximation for the backward pass of the algorithm. While Defense-GAN is also supposed to be vulnerable against [C], as it must also obfuscate gradients, [C] reported a failure of attacking Defense-GAN. This motivated us to focus on approaches based on generative models than any other approaches including denoising.
>
> The work of [A] is an interesting approach to defend against a universal adversarial perturbation, even though it does not show its performance against an adversary who attacks each data point differently. However, assuming a white-box adversary, this can also be considered as an input transformation, so it is very likely to suffer performance drop via differentiable approximation of denoising mechanism.
>
> The work of [B] is very impressive and attacking it would be an interesting challenge. However, we are a bit reluctant to relate this to the manifold assumption yet, due to the lack of knowledge about how the image of data manifold would look like in the intermediary feature space. Since they used adversarial training for both the baseline models and their feature denoising models, the approach of [C] will not be a direct threat to their model as the author of [C] admitted that adversarial training does not seem to cause obfuscated gradients. However, the benefit by introducing feature denoising blocks may disappear after the method of [C] is applied.
>
> Reference:
> [A] Defensive Denoising Methods Against Adversarial Attack
> [B] Feature Denoising for Improving Adversarial Robustness
> [C] Obfuscated Gradients Give a False Sense of Security: Circumventing Defenses to Adversarial Examples
> [D] Feature Squeezing: Detecting Adversarial Examples in Deep Neural Networks
> [E] MagNet: a Two-Pronged Defense Against Adversarial Examples
> [F] Countering Adversarial Images Using Input Transformations
> [G] Bypassing Feature Squeezing by Increasing Adversary Strength
> [H] MagNet and "Efficient Defenses Against Adversarial Attacks" are Not Robust to Adversarial Examples

---

### Author Response · Authors · 2019-11-15
**Minor changes made after rebuttal**

We found the following minor typo after posting the rebuttal, and it is now corrected.
 - The loss function is corrected from "likelihood" to "negative log-likelihood".
 - The title of Figure 7a is corrected from "Isotropic Gaussian" to "Mixture of 2 Gaussians".

---

### Decision · Program_Chairs · 2019-12-19

**Decision:**

Accept (Poster)

**Comment:**

This paper studies the role of topology in designing adversarial defenses. Specifically , the authors study defense strategies that rely on the assumption that data lies on a low-dimensional manifold, and show theoretical and empirical evidence that such defenses need to build a topological understanding of the data.

Reviewers were initially positive, but had some concerns pertaining to clarity and limited experimental setup. After a productive rebuttal phase, now reviewers are mostly in favor of acceptance, thanks to the improved readibility and clarity. Despite the small-scale experimental validation, ultimately both reviewers and AC conclude this paper is worthy of publication.